

# Nitrogen and oxygen availabilities control water column nitrous oxide production during seasonal anoxia in the Chesapeake Bay

Qixing Ji[1], Claudia Frey[1], Xin Sun[1], Melanie Jackson[2], Yea-Shine Lee[1], Amal Jayakumar[1], Jeffrey C. Cornwell[2] and Bess B. Ward[1]

[1]Department of Geosciences. Princeton University, Princeton, 08544, New Jersey, USA
[2]Horn Point Laboratory, University of Maryland Center for Environmental Science, Cambridge, 21613, Maryland, USA

*Correspondence to*: Qixing Ji (qji@princeton.edu)

**Abstract.** Nitrous oxide ($N_2O$) is a greenhouse gas and an ozone depletion agent. One of the major uncertainties in the global $N_2O$ budget is the contribution of the coastal region, including estuaries, which can be sites of intense $N_2O$ efflux. Incubation experiments with nitrogen stable isotope tracer ($^{15}N$) enabled the investigation of the environmental controls of $N_2O$ production in the water column of Chesapeake Bay, the largest estuary in North America. The highest potential rates of $N_2O$ production ($7.5 \pm 1.2$ nmol-N $L^{-1}$ $hr^{-1}$) were detected during summer anoxia, during which oxidized nitrogen species (nitrate and nitrite) were absent from the water column. At the top of the anoxic layer, $N_2O$ production from denitrification was stimulated by addition of nitrate and nitrite. The relative contribution of nitrate and nitrite to $N_2O$ production was positively correlated with the ratio of nitrate to nitrite concentrations. Increased oxygen availability, up to 7 μM oxygen inhibited both $N_2O$ production and the reduction of nitrate to nitrite. Therefore, reducing the nitrogen input into the Chesapeake Bay has two potential impacts on the $N_2O$ efflux: In the short-term, $N_2O$ emission will be mitigated due to nitrogen deficiency. In the long-run, eutrophication will be alleviated and subsequent re-oxygenation of the bay will further inhibit $N_2O$ production.

## 1 Introduction

Nitrous oxide ($N_2O$) is a strong greenhouse gas with 298-fold higher global warming potential per mole than that of carbon dioxide. $N_2O$ is also a catalyst of ozone depletion in the stratosphere. Since the Industrial Revolution, the $N_2O$ atmospheric concentration has been increasing at an unprecedented rate, and the current concentration is the highest in the last 800,000 years of Earth's history (Schilt et al., 2010). The contribution of $N_2O$ emissions to global warming and ozone depletion will increase because $N_2O$ is not as strictly regulated as are $CO_2$ and halocarbon compounds. With the successful mitigation of halocarbon compounds accomplished by the Montreal Protocol, $N_2O$ is likely to be the single most important anthropogenically emitted ozone-depleting agent in the 21st century (Ravishankara et al., 2009).



Microbial processes are responsible for the majority of $N_2O$ production, both in natural and anthropogenically impacted environments. These pathways include oxidative and reductive processes occurring at the full range of environmental oxygen concentrations. In the presence of oxygen, $N_2O$ can be produced as a by-product during autotrophic aerobic ammonium ($NH_4^+$) oxidation to nitrite ($NO_2^-$) by bacteria (Arp and Stein, 2003) and archaea (Santoro et al., 2011). The

production of $N_2O$ can also occur via $NO_2^-$ reduction by nitrifying organisms, termed nitrifier denitrification. This process was demonstrated in cultures (Poth and Focht, 1985; Frame and Casciotti, 2010), and in the water column of the subtropical North Pacific Ocean (Wilson et al., 2014). Under low oxygen and anoxic conditions, $N_2O$ is produced via stepwise, enzyme-mediated heterotrophic denitrification, i.e. the reduction of nitrate ($NO_3^-$) and $NO_2^-$, with organic matter as the electron donor. $N_2O$ is not produced via anaerobic ammonium oxidation (anammox), another important nitrogen removal process in

the natural environment (Kartal et al., 2011).

      The increase of atmospheric $N_2O$ is attributed to intensification of human activities (e.g. fossil fuel combustion, fertilizer application, human and animal waste disposal), which alter the microbial nitrogen cycle in the biosphere. Increased nitrogen supply from fertilizer and atmospheric deposition causes increased $N_2O$ emission not only from agricultural land, but also in rivers, streams and coastal waters (Ciais et al., 2013; Thompson et al., 2014). Among these aquatic environments,

the most intense $N_2O$ efflux originates from estuaries and associated river networks, which occupy 0.3% of global waters (Dürr et al., 2011) but contribute up to 10 % of anthropogenic fluxes (Seitzinger and Kroeze, 1998; Ciais et al., 2013). Being the largest estuary in the North America, the Chesapeake Bay and its tributaries have been identified as a $N_2O$ source in the summertime (June to September), during which surface $N_2O$ oversaturation with respect to air occurs (Elkins et al., 1978; Kaplan et al., 1978; McElroy et al., 1978). The water column is characterized by strong oxygen gradients (equilibrium

with atmosphere at the surface and complete anoxia below ~ 10 m), depletion of oxidized nitrogen species ($NO_3^-$ and $NO_2^-$), and accumulation of ammonium in the deep water (Lee et al., 2015b). Since the 18[th] century, increased population, expansion of industrialization and land use changes in the Northeastern US have increased nutrient input into the Chesapeake tributaries and caused expansion of summertime anoxia (Cooper and Brush, 1993; Boesch et al., 2001). Increased microbial activities driving carbon assimilation and respiration have been demonstrated in the vicinity of the oxic-

anoxic interface in the water column (Lee et al., 2015a). The global estimate of estuarine $N_2O$ fluxes is poorly constrained,

partly because of the paucity of data on N₂O production and the associated environmental controlling factors in estuarine systems such as Chesapeake Bay.

Here we report a pilot study using nitrogen stable isotope incubation experiments to investigate N₂O production in Chesapeake Bay, and to quantify its dependence on the availabilities of oxygen and oxidized nitrogen. Because seasonal anoxia occurs at the study site in the central region of the Chesapeake Bay, reductive pathways of N₂O production are the main focus. Further understanding of the environmental controls on N₂O production in estuaries will facilitate the design of effective environmental engineering projects to mitigate N₂O emission.

## 2 Methods

### 2.1 Sample acquisition and processing

Sampling and incubation experiments were carried out on July 19, 2016, November 17, 2016 and May 3, 2017, corresponding to typical conditions of summer, autumn and spring, respectively. Samples were collected at 38.55 ˚N, 76.43 ˚W (bottom depth 26.5 m) close to the mouth of the Choptank River in the central region of the Chesapeake Bay. Conductivity-temperature-depth and oxygen were measured with a YSI sonde package (Model 600XLM with a 650 MDS display logger) equipped with a diaphragm pump which was deployed for water sampling. The oxygen sensor had a detection limit of ~ 5 μmol L$^{-1}$. Samples for $NO_2^-$ and $NO_3^-$ concentration measurements were filtered (0.22 μm poresize, Sterivex-GP, EMD Millipore) and frozen at -80 ˚C until analysis. Discrete samples for N₂O concentration were collected directly from the pump outlet into the bottom of acid washed, 60 mL glass serum bottles (Catalog # 223745, Wheaton, Millville, NJ). Bottles were sealed with butyl rubber stoppers (Catalog # W224100-202, Wheaton, Millville, NJ) and aluminium rings while submerged under water pumped from depth to avoid atmospheric N₂O and oxygen contamination. Samples for characterizing N₂O concentration profile were preserved immediately after filling by injecting 0.1 mL saturated $HgCl_2$. Samples for N₂O incubation experiments (section 2.2) were acquired from 12 m, 17 m and 19 m during July 2016, November 2016 and May 2017, respectively, and sealed the same way as described above for discrete N₂O concentration samples, and stored in the dark at 4 ˚C without adding $HgCl_2$. Samples for denitrifying *nirS* gene abundance were collected at



14, 17 and 19 m by filtering 600mL - 2000mL of water through 0.22 μm filter (Sterivex-GP, EMD Millipore) and frozen at -80 ℃ until DNA extraction and analysis.

Samples for total dissolved inorganic carbon (DIC=$[H_2CO_3]$+$[HCO_3^-]$+$[CO_3^{2-}]$) and community respiration rates were collected only in July 2016. The DIC samples were preserved with mercuric chloride ($HgCl_2$) for initial conditions, while

biochemical oxygen demand (BOD) bottles were incubated in a temperature-controlled environmental chamber (±1 ℃ of in situ water temperatures). After 24 h, samples were siphoned from the vials, preserved with $HgCl_2$, and respiration rates were determined as the difference in DIC between initial and final samples divided by the 24 hours (Lee et al., 2015b).

## 2.2 $^{15}$N incubation experiments for $N_2O$ production

Within 3 hours of sampling, incubation experiments were initiated at the Horn Point Laboratory, Cambridge,

Maryland. Samples were divided into two sets for nitrogen and oxygen manipulation experiments.

Dissolved inorganic nitrogen (DIN) manipulation: The nitrogen manipulation experiment was conducted only in July 2016 because $NO_2^-$ and $NO_3^-$ were absent from the water column (see section 3.1). A small (3 ml) headspace was created in the serum bottles, which were subsequently flushed with helium for 10 minutes to minimize oxygen contamination from sampling and transportation. In order to detect $N_2O$ production, ~1.2 nmol $N_2O$ was injected to each bottle, reaching a

concentration of ~20 nmol $L^{-1}$ in the water phase (calculated equilibrium concentration (Weiss and Price, 1980) with 3 mL headspace and 57 mL water). Two suites of $^{15}$N tracer solutions ($^{15}NO_2^-$ plus $^{14}NO_3^-$, $^{15}NO_3^-$ plus $^{14}NO_2^-$, 0.1 mL of total volume of tracer addition) were injected to designated bottles to achieve ratios of $NO_2^-$ : $NO_3^- \approx$ 1:10, 1:3, 3:1 and 10:1, with $^{15}$N fraction labelled between 0.016 and 0.16 (Table 1, experiment 2-A to 2-H). This allows simultaneous detection of $N_2O$ production from $NO_2^-$ and $NO_3^-$ at different ratios of $NO_2^-$ to $NO_3^-$ concentration. Tracer solutions were made from deionized

water, and were flushed with helium prior to addition to incubation experiments. Initial conditions (one bottle of each time courses) were sampled within 30 min of tracer addition by injecting 0.1 mL saturated $HgCl_2$. Incubations lasted ~2 hours at a temperature difference < 0.5 °C of those of *in situ*, during which duplicate bottles were preserved with $HgCl_2$ every 40 to 60 minutes, totalling seven bottles over four time points, including the initial.

Oxygen manipulation: The oxygen manipulation experiment was conducted in July 2016 and November 2016.

Headspace (3 – 8 ml) was created before flushing with helium for 10 minutes. Oxygen-saturated site water was made by air-

equilibration at *in situ* temperature. To achieve different oxygen levels, 0.2, 0.5, 1.0, 2.0 or 5.0 ml of oxygen-saturated site water was injected. With a final volume of ~3 mL of headspace during the course of the incubation, the oxygen concentrations in the water phase were 0.3 to 6.4 μmol L$^{-1}$ in July 2016 (Table 1, experiment 3-A – 3-J), and were 0.2 to 7.3 μmol L$^{-1}$ in November 2016 (Table 1, experiment 5-A – 5-J) after the calculated equilibration between headspace and

seawater (Garcia and Gordon, 1992). The control experiment was designated as anoxic with no oxygen addition (Table 1, experiment 1-A and 1-B, 4-A and 4-B). After oxygen adjustment, ~1.2 nmol N$_2$O was injected into each bottle, and two suites of $^{15}$N tracer solutions ($^{15}$NO$_2^-$ plus $^{14}$NO$_3^-$, $^{15}$NO$_3^-$ plus $^{14}$NO$_2^-$, 0.1mL) were injected to achieve final concentration of 5 μmol L$^{-1}$ NO$_2^-$ and NO$_3^-$. The $^{15}$N fraction for NO$_2^-$ or NO$_3^-$ during the incubation experiments are shown in Table 1.

### 2.3 Analytical procedures

For water column nutrients, dissolved NO$_2^-$ was measured using a colorimetric method (Hansen and Koroleff, 2007) and NO$_3^-$ + NO$_2^-$ was measured using a hot (90 ℃) acidified vanadium (III) reduction column coupled to a chemiluminescence NO/NOx Analyzer (Teledyne API, San Diego, CA)  (Garside, 1982; Braman and Hendrix, 1989). DIC was measured with an automated infrared analyzer (Apollo SciTech, Newark, DE) as previously reported (Lee et al., 2015b). Preserved N$_2$O samples were stored in the dark at room temperature (~22 ℃) for less than three weeks before analysis.

Dissolved N$_2$O was extracted by flushing with helium for 40 min at a rate of 37 ml min$^{-1}$ (extraction efficiency 99 ± 2 %), and subsequently cryo-trapped by liquid nitrogen and isolated from interfering compounds (H$_2$O, CO$_2$) by gas chromatography (Weigand et al., 2016). Pulses of purified N$_2$O were injected into a Delta V$^{Plus}$ mass spectrometer (Thermo Fisher Scientific, Waltham, MA) for mass (m/z = 44, 45, 46) and isotope ratio (m$_1$/m$_2$ = 45/44, 46/44) measurements. The amount of N$_2$O was calibrated with standard N$_2$O vials, which were made by injecting 1, 2, or 5 nmol N$_2$O-N into 20 mL

glass vials (Catalog # C4020-25, Thermo Fisher Scientific, Waltham, MA).

After N$_2$O analysis, samples incubated with $^{15}$NO$_3^-$ were also assayed for $^{15}$NO$_2^-$ to determine rates of NO$_3^-$ reduction. Two millilitres of each sample were transferred from the 60-mL serum bottle to a 20-mL glass vial and then flushed with helium for 10 min. Dissolved $^{15}$NO$_2^-$ was converted to N$_2$O using the acetic acid-treated sodium azide solution for quantitative conversion (McIlvin and Altabet, 2005). Resulting N$_2$O was measured on the Delta V$^{Plus}$ for nitrogen isotope

ratio so as to determine the $^{15}$N enrichment of NO$_2^-$.





For molecular analysis, DNA extraction and qPCR for the *nirS* gene using SYBR Green were performed as previously

described (Jayakumar et al. (2009); 2013). Extracted DNA was quantified using PicoGreen fluorescence (Molecular Probes,

Eugene, OR) prior to the qPCR assay. Samples for qPCR were run in triplicates including a no template control, a no Primer

control and 5 different dilutions of a *nirS* standard. Threshold cycle (Ct) values were obtained using automatic analysis

settings of the quantitative PCR and further used to calculate the gene copy numbers as described in Jayakumar et al. (2013).

## 2.4 Data analysis

N$_2$O concentration was calculated from the amount of N$_2$O detected by mass spectrometry divided by the volume of

water in the serum bottles. N$_2$O production ($R$) was calculated from the progressive increase in $^{45}$N$_2$O and $^{46}$N$_2$O

concentrations in each serum bottle over the time course experiments.

$$R = \frac{1}{F} \times \left( \frac{d^{45}N_2O}{dt} + 2 \times \frac{d^{46}N_2O}{dt} \right)$$

(1)

where $d^{45}N_2O/dt$ and $d^{46}N_2O/dt$ represent the production rates (nmol-N L$^{-1}$ hr$^{-1}$) of mass 45 and 46 N$_2$O during incubation. F

represents the $^{15}$N fraction in the initial substrate (NO$_2^-$ or NO$_3^-$). Rates were considered significant based on the linear

regression of the time course data ($p < 0.05$, n=7, student t-test). The detection limit for N$_2$O production is 0.002 nmol-N L$^{-1}$

hr$^{-1}$.

The rate of NO$_3^-$ reduction to NO$_2^-$ was calculated as

NO$_2^-$ production = ($d^{15}$NO$_2^-/dt$ ) / $F$            (2)

where $d^{15}NO_2^-/dt$ represents the production rate of $^{15}$NO$_2^-$ (nmol-N L$^{-1}$ hr$^{-1}$), which is calculated as the slope of $^{15}$NO$_2^-$

concentrations versus time. $F$ represents initial substrate $^{15}$NO$_3^-$ enrichment. Rates were considered significant based on

linear regression of the time course data (p<0.05, student's t-test). The detection limit for NO$_2^-$ production is 0.05 nmol-N L$^{-1}$

20   hr$^{-1}$.



## 3 Results and discussion

### 3.1 Water column features

The physical and chemical properties of the water column in central Chesapeake Bay experience seasonal variation (Fig. 1). Temperature and salinity differed among the three seasons but were essentially constant in the top 7 m of the water column on the three sampling dates. In July, the water column was stratified because of lower salinity (~ 16 PSU) and higher temperature (~ 28.5 ℃) in the top ~ 10 m resulting in a pronounced halocline and thermocline (Fig. 1a and 1b). Less pronounced stratification in May and November was due to weaker temperature difference between top 10 m and below. The July oxygen profile shows significant concentration decrease between 3 to 10 m (Fig. 1c), with a sharp oxycline (~ 30 μmol $L^{-1}$ $m^{-1}$). Below 10 m, the oxygen concentration was below detection of the sensor (~ 5 μmol $L^{-1}$) and was likely anoxic. However, sulphide compounds were most likely not present in July at depth; the water samples were free of any hydrogen sulphide odour. No anoxic layer was observed in May and November (Fig. 1c), and previous studies showed that the water column of the Chesapeake Bay was reoxygenated following summertime anoxia during winter and spring (Lee et al., 2015a).

In July, $N_2O$ concentration was close to air-saturation level (6.6 nmol $L^{-1}$) at the surface layer (Fig. 1d). In the low oxygen layer (below 12 m), $N_2O$ was undersaturated (2.0 – 3.7 nmol $L^{-1}$, 20 – 50 % air-saturation). This was the only instance of $N_2O$ undersaturation observed in three sampling trips; $N_2O$ concentrations in May were constant at air-saturation level of 11.2 nmol $L^{-1}$ between 3 and 17 m in the water column; in November, the $N_2O$ concentrations varied between 9.8 and 11.2 nmol $L^{-1}$. The concentrations of $NO_3^-$ and $NO_2^-$ (Fig. 1d and 1e) in July were below 0.02 μmol $L^{-1}$ within the sampling depth interval (top 17 m of water column). Measureable levels of oxidized nitrogen species were found in May and November. The concentrations of $NO_2^-$ and $NO_3^-$ in May were 20 and 0.5 μmol $L^{-1}$, respectively; and the concentrations decreased with depth. In November, $NO_3^-$ and $NO_2^-$ were depleted at the surface (~ 3 m) and their concentrations increased with depth; at 17 m the concentrations of $NO_3^-$ and $NO_2^-$ were 5.0 and 0.4 μmol $L^{-1}$, respectively.

As a proxy for the size of the denitrifying community, the abundance of the *nirS* gene was (5.91 ± 0.1) × $10^4$ copy $mL^{-1}$ at 14 m in July, which was the highest among the three sampling trips (Fig. 1g). Lowest *nirS* gene abundance (9.1 ± 1.3) × $10^3$ copy $mL^{-1}$ was observed in May at 19 m. The abundance of *nirS* was measured only at the depths at which incubations were performed, and the *nirS* abundance positively correlated with measured rates of $N_2O$ production (see section 3.2). In




July 2016, water column DIC concentrations ranged from 1,377 to 1,831 µmol L$^{-1}$, with the highest concentrations below 10

m. Average community respiration rates at 3 m and 14 m depth were 2.01 and 0.63 µmol L$^{-1}$ hr$^{-1}$, respectively.

## 3.2 Active N$_2$O production by denitrification

Active N$_2$O production was detected (Fig. 2) in the control experiment (helium-flushed anoxic incubation) at the top of

anoxic layer (~ 12.3 m) in July 2016; rates of N$_2$O production from NO$_2^-$ and NO$_3^-$ reduction were 5.42±0.35 and 2.04±0.86

nmol-N L$^{-1}$ hr$^{-1}$, respectively, when 5 µmol L$^{-1}$ NO$_2^-$ or NO$_3^-$ was added. In November 2016, the water column was

oxygenated (> 180 µmol L$^{-1}$), and the rates of N$_2$O production from NO$_2^-$ and NO$_3^-$ reduction at 17 m in the anoxic control

(helium-flushed anoxic incubation) were 0.33±0.01 and 0.95±0.35 nmol-N L$^{-1}$ hr$^{-1}$, respectively. In May 2017, no N$_2$O

production was detected.

The total N$_2$O production rate of 7.5±1.2 nmol-N L$^{-1}$ hr$^{-1}$ in July 2016 is lower than the measurements (18 – 77 nmol-N

L$^{-1}$ hr$^{-1}$) made 40 years ago in the Potomac River (McElroy et al., 1978), a tributary to the Chesapeake Bay. This difference

could be due to much higher water column nutrients in the Potomac River (NO$_2^-$ plus NO$_3^-$ concentration > 30 µmol L$^{-1}$) at

that time, and presumably denser microbial populations because of sediment resuspension (4 – 10 m water depth). With

added substrates (NO$_2^-$ and NO$_3^-$) being more than an order of magnitude higher than *in situ* levels in July 2016, and the

anoxic conditions being used in the November 2016 experiments (in situ [O$_2$] > 180 µmol L$^{-1}$), N$_2$O production rates

reported here are potential rates, which nevertheless highlight the potential for N$_2$O production in anoxic waters responding

rapidly (within hours) to pulses of oxidized nitrogen.

Based on the *nirS* gene abundance, the denitrifying population was more abundant in July (summer) than November

(autumn), and was the smallest in May (spring) in the lower water column (14 – 19 m) of the Chesapeake Bay (Fig. 1g). In

July highest N$_2$O production rates co-occurred with the highest *nirS* abundances (Fig. 2). While the water column oxygen

was > 180 µmol L$^{-1}$ in November, the *nirS* gene abundance supported potential denitrification at a N$_2$O production rate of

1.28 ± 0.35 nmol-N L$^{-1}$ hr$^{-1}$ in anoxic incubation experiments. In May when hypoxic conditions had not yet developed, no

N$_2$O production was detected, and the *nirS* abundance (9.1 ×10$^3$ copies mL$^{-1}$) was the lowest among three sample dates. This

pattern is consistent with a metatranscriptome analysis that showed lowest transcript ratios for denitrification in June before

the onset of hypoxia and highest ratios in August when anoxia was most pronounced (Eggleston et al., 2015).

Denitrification, as a major pathway of fixed nitrogen removal, is critical to mitigating eutrophication in natural waters. In spring, runoff from the anthropogenically influenced watershed results in high $NO_3^-$ and $NO_2^-$ concentrations in the Bay. The subsequent increase in denitrification activity, which peaks in summertime, depletes water column $NO_3^-$ and $NO_2^-$ (Baird et al., 1995; Boynton et al., 1995). Even when the substrates $NO_2^-$ and $NO_3^-$ were nearly absent in the summertime,

the water column was readily capable of denitrification. The net $N_2O$ production rates could serve as a proxy for estimating nitrogen loss. It is estimated that 1% of total denitrified nitrogen is converted to $N_2O$ in river networks (Beaulieu et al., 2011) so the ratio of $N_2O : N_2$ during denitrification = 1 : 100. Assuming that $N_2O$ production occurs at a rate of 7 nmol-N $L^{-1}$ $hr^{-1}$ within 0.2 m of the oxic-anoxic interface in summertime (based on the July 2016 control data, $N_2O$ production from $NO_3^-$ plus $NO_2^-$), denitrification yields a potential water column N removal rate of 140 µmol-N $m^{-2}$ $hr^{-1}$, or 0.24 mg-N $m^{-2}$ $d^{-1}$. In

addition, the sediment in the Bay is capable of anaerobic ammonia oxidation (Rich et al., 2008) and denitrification (Kemp et al., 1990; Kana et al., 2006). Total sedimentary $N_2$ production, measured by the acetylene block reduction method (Kemp et al., 1990) and $N_2$ accumulation method (Kana et al., 2006) recorded areal rates of 50 – 70 µmol-N $m^{-2}$ $hr^{-1}$. Therefore, the sediment-water system in the Chesapeake Bay is effective in biological nitrogen removal.

### 3.3 $N_2O$ production pathways regulated by availability of nitrogen substrate

The ratio of the rates of $N_2O$ production from $NO_2^-$ reduction vs. $N_2O$ production from $NO_3^-$ reduction positively correlates with the ratio of $NO_2^- : NO_3^-$ concentrations (Fig. 3). This suggests increasing $NO_2^-$ ($NO_3^-$) availability favours $N_2O$ production from $NO_2^-$ ($NO_3^-$) reduction. At concentration ratios of $NO_2^- : NO_3^- < 0.5$, the ratios of rates were similar to the concentration ratio, 0.3±0.2. At a concentration ratio of $NO_2^- : NO_3^- = 1 : 1$, the ratio of rates of $N_2O$ production from respective substrates measured from replicate experiments varied from 0.6 to 2.6. At $NO_2^- : NO_3^- = 10$, the ratio of rates was

greater than 10. Therefore, the primary nitrogen source of $N_2O$ production via denitrification depends in part on the relative availability of the substrate ($NO_2^-$ or $NO_3^-$). The following discussion is based on data from July 2016 because this was the only instance on which the DIN concentration ratio experiment was conducted.

Denitrification is a step-wise enzymatic reduction from $NO_3^-$, $NO_2^-$, NO, $N_2O$ to $N_2$. However, the pathway is somewhat modular (Graf et al., 2014), i.e., many organisms possess only one or a few steps, rather than the complete





pathway. In complete denitrifiers (organisms capable of reducing $NO_3^-$ to $N_2$), the degree to which intermediates (i.e. $NO_2^-$) exchange across cellular membranes with the ambient environment is unknown (Moir and Wood, 2001). To estimate the exchange of intracellular and ambient $NO_2^-$ during $NO_3^-$ reduction to $N_2O$ by denitrifiers, the following calculations use the conditions and results from experiment 2-H (Table 1) because this experiment had the highest ambient $NO_2^-$ pool and an

exchange between the pools could be easily detected. During $NO_3^-$ reduction to $N_2O$, if denitrifiers reduce $^{15}NO_3^-$ (total 1.2 μmol L$^{-1}$, $^{15}N$ fraction labeled 0.16) to $^{15}NO_2^-$ at maximal rate (0.2 μmol-N L$^{-1}$ hr$^{-1}$, see section 3.4) and the product fully exchanges with the ambient $^{14}NO_2^-$ (10 μmol L$^{-1}$, $^{15}N$ fraction labeled 0.0037), after 2 hours, the $^{15}N$ addition to the total $NO_2^-$ pool will be 0.0064 μmol L$^{-1}$:

(Rate of $NO_2^-$ production from $NO_3^-$ × incubation time × initial fraction labelled of $NO_3^-$ × concentration of $NO_3^-$) /

(concentration of $NO_2^-$)

= (0.2 μmol-N L$^{-1}$ hr$^{-1}$ × 2 hr × 0.16 × 1 μmol-N L$^{-1}$) / (10 μmol-N L$^{-1}$) = 0.0064 μmol L$^{-1}$,

and the resulting $^{15}N$ fraction (unitless) of $NO_2^-$ will be 0.004:

($^{15}N$ addition to $NO_2^-$ + initial fraction labelled of $NO_2^-$ × initial concentration of $NO_2^-$) / (total concentration of $NO_2^-$)

= (0.0064 μmol L$^{-1}$ + 0.0037 × 10 μmol L$^{-1}$) / (10 + 0.0064) μmol L$^{-1}$ ≈ 0.004.

Assuming 10 nmol-N L$^{-1}$ hr$^{-1}$ as the rate of $N_2O$ production from $NO_2^-$ reduction (twice as high as the $NO_2^-$ → $N_2O$ rate shown in fig. 3; $^{15}N$ fraction labeled of $NO_2^-$ = 0.004), and the initial $N_2O$ concentration as 20 nmol L$^{-1}$ (described in section 2.2; $^{15}N$ fraction labeled of $N_2O$ = 0.0037), after 2 hours, the resulting $^{15}N$ fraction of $N_2O$ will be 0.0038:

(($^{15}N$ fraction labelled of $NO_2^-$ × rate of $N_2O$ production from $NO_2^-$ × incubation time) + (initial fraction labelled of $N_2O$ × initial concentration of $N_2O$ × molar nitrogen in molar $N_2O$)) / ((rate of $N_2O$ production from $NO_2^-$ × incubation time) +

(initial concentration of $N_2O$ × molar nitrogen in molar $N_2O$))

= ((0.004 × 10 nmol-N L$^{-1}$ hr$^{-1}$ × 2 hr) + (0.0037 × 20 nmol-$N_2O$ L$^{-1}$ × 2N/$N_2O$)) / (10 × 2 + 20 × 2) nmol-N L$^{-1}$ = 0.0038

The calculated $^{15}N$ fraction of $N_2O$ (0.0038) is much lower than the measured $^{15}N$ fraction of $N_2O$ (> 0.02) in experiment 2H. This means that full exchange of $NO_2^-$ during $NO_3^-$ reduction to $N_2O$, at maximum possible rates of $NO_3^-$ reduction to $NO_2^-$ and $N_2O$, would yield a rate of $N_2O$ production from $NO_3^-$ much lower than observed in the experimental results. Thus, we

concluded that the exchange between intracellular and ambient $NO_2^-$ during $NO_3^-$ reduction to $N_2O$ by the denitrifying community in Chesapeake Bay is limited. Such a tight coupling among nitrate reduction, nitrite reduction and nitric oxide reduction suggests the co-occurrence of the respective functional genes and enzymes in the cell of nitrate reducers. Both dissimilatory nitrate and nitrite reducers are able to produce $N_2O$ independently, so total $N_2O$ production can be quantified accurately by separate measurement of $NO_3^-$ and $NO_2^-$ reduction.

### 3.4 Oxygen inhibits $N_2O$ production by denitrification

Oxygen availability may mediate the denitrification response to DIN availability. The incubation experiments demonstrated that potential $N_2O$ production was initiated when external nitrogen sources were added. Therefore, controlling the influx of nitrogen into Chesapeake Bay could mitigate the efflux of $N_2O$ and its environmental and climate impacts. Since the late 20[th] century, Chesapeake Bay has received increased anthropogenic nitrogen loading from various sources including fertilizer (Groffman et al., 2009), untreated sewage (Kaplan et al., 1978) and atmospheric deposition (Russell et al.,

1998; Loughner et al., 2016). The Chesapeake Bay was identified in 1978 as a potential $N_2O$ source due to $N_2O$ supersaturation at the surface (Elkins et al., 1978). Since then, measures have been successfully enforced to control the nitrogen runoff into the bay from the tributaries (Boesch et al., 2001; Program, 2017). The near absence of summertime water column $NO_2^-$ + $NO_3^-$ concentrations near the middle of Chesapeake Bay as shown in this study and others (Lee et al., 2015a) could prevent $N_2O$ emission. Contrary to the studies conducted in the 1970s (Elkins et al., 1978; Kaplan et al., 1978;

McElroy et al., 1978; Elkins et al., 1981), our measurements from July 2016 showed surface $N_2O$ concentration was close to air-saturation, and undersaturation of $N_2O$ within the anoxic layer (Fig. 1d). Assuming $N_2O$ concentration was in steady state, water column $N_2O$ undersaturation is a sign of $N_2O$ consumption, which lowers $N_2O$ flux from the Chesapeake Bay and is an indication of $N_2O$ serving as an electron acceptor during organic matter remineralization. However, $N_2O$ consumption is inhibited by trace amounts of oxygen, and is thus confined within the anoxic layer; the Chesapeake Bay is



unlikely to be a sink for atmospheric $N_2O$ because the downward mixing and molecular diffusion introduce oxygen to the anoxic layer, inhibiting $N_2O$ consumption.

The sensitivities to increasing $[O_2]$ of $NO_2^-$ reduction and $NO_3^-$ reduction to $N_2O$ were evaluated in samples from July and November 2016 (Fig. 4). The control experiment (anoxic incubation, see Section 3.2) showed a total $N_2O$ production

rate (from $NO_2^-$ plus $NO_3^-$ reduction) of 7.5±1.2 and 1.28 ± 0.35 nmol-N $L^{-1}$ $hr^{-1}$ during July 2016 and November 2016, respectively. Increasing $[O_2]$ generally decreased $N_2O$ production rates from denitrification. In July 2016, under $[O_2]$ = 0.3 $\mu$mol $L^{-1}$, $N_2O$ production from $NO_2^-$ reduction decreased from 5.4 to 2.5 nmol-N $L^{-1}$ $hr^{-1}$, whereas the rate of $NO_3^-$ reduction to $N_2O$ increased from 2.0 to 3.5 nmol-N $L^{-1}$ $hr^{-1}$. Further increase in $[O_2]$, up to 6.4 $\mu$mol $L^{-1}$, significantly inhibited the rate of $N_2O$ production from both $NO_2^-$ and $NO_3^-$ reduction (Fig. 4a). Note that 6 $\mu$mol $L^{-1}$ $[O_2]$ did not fully inhibit $N_2O$

production from $NO_2^-$ reduction, the rate of which was 0.08 nmol-N $L^{-1}$ $hr^{-1}$. However, $N_2O$ production from $NO_3^-$ reduction was completely inhibited when $[O_2]$ > 0.6 $\mu$mol $L^{-1}$. Similar to results from July 2016, in November 2016, increasing $[O_2]$ gradually decreased rates of $NO_2^-$ reduction to $N_2O$; no rates were detected when $[O_2]$ > 2 $\mu$mol $L^{-1}$. Rates of $NO_3^-$ reduction to $N_2O$ were not detected at $[O_2]$ > 0 $\mu$mol $L^{-1}$ (Fig. 4b). A previous study found that $NO_3^-$ reduction to $N_2O$ was less oxygen sensitive than $NO_2^-$ reduction to $N_2O$ in open ocean oxygen minimum zones (Ji et al., 2015). The reasons for the opposite

behavior in Chesapeake Bay are unknown.

Rate of $NO_3^-$ reduction to $NO_2^-$ was also measured in July 2016 to supplement the sensitivity analysis of denitrification to oxygen. The rate of $NO_3^-$ reduction to $NO_2^-$ was 100 nmol $L^{-1}$ $hr^{-1}$ under anoxic condition. At $[O_2]$ = 0.3 $\mu$mol $L^{-1}$, the rate doubled, to 200 nmol-N $L^{-1}$ $hr^{-1}$ (Fig. 4). Further increase of $[O_2]$ significantly decreased the rate of $NO_3^-$ reduction to $NO_2^-$. However, at $[O_2]$ = 6.4 $\mu$mol $L^{-1}$ $NO_3^-$ reduction to $NO_2^-$ was still detectable at 0.82 ±0.06 nmol-N $L^{-1}$ $hr^{-1}$ (Fig. 5).

These results suggest that the oxic-anoxic interface in the water column is potentially a "hot spot" for $N_2O$ production from denitrification, and that oxygenation of the water column in the Chesapeake Bay, even micro-molar level oxygen, would significantly mitigate $N_2O$ production. Both July 2016 and November 2016 data showed the difference in the effect of oxygen on $N_2O$ production from $NO_2^-$ vs. $NO_3^-$ reduction. Samples from July 2016 showed 98% and complete inhibition on $N_2O$ production from $NO_2^-$ and $NO_3^-$ reduction at $[O_2]$ = 6 $\mu$mol $L^{-1}$, respectively. The November 2016 samples showed 94



% and complete inhibition on $N_2O$ production from $NO_2^-$ and $NO_3^-$ reduction at $[O_2] = 0.4$ μmol $L^{-1}$, respectively. These results can be explained by the differences in physiology among microbial communities mediating these processes. Both nitrifiers and denitrifiers are present in the Chesapeake Bay (Bouskill et al., 2012; Hong et al., 2014) and they are capable of $NO_2^-$ reduction to $N_2O$, whereas $NO_3^-$ reduction to $N_2O$ is solely mediated by denitrifiers. Nitrifier denitrification is an important $N_2O$ production pathway occurring under the full range of oxygen environments in agricultural soil (Zhu et al., 2013) and the open ocean (Wilson et al., 2014). Partial denitrification ($NO_3^-$ reduction to $N_2O$) however, is moderately oxygen sensitive. Thus, increasing oxygen inhibits the activities of denitrifiers, as demonstrated in decreasing rates of $NO_3^-$ reduction to $N_2O$ (Fig. 3) and $NO_3^-$ reduction to $NO_2^-$ (Fig. 5). Increasing oxygen does not completely inhibit $N_2O$ production activity of nitrifiers but probably lowers the $N_2O$ production rates by nitrifier denitrification (Zhu et al., 2013).

Nitrification is a possible pathway for $N_2O$ production within the sharp oxycline of the Chesapeake Bay water column. $N_2O$ is produced as a byproduct via aerobic ammonium oxidation under low oxygen conditions (Anderson, 1964). The yield of $N_2O$ (molar ratio of $N_2O$ production to ammonium oxidation) increases with decreasing oxygen (Goreau et al., 1980). Culture (Qin et al., 2017) and field studies (Bristow et al., 2016; Peng et al., 2016) have shown high affinity of oxygen (< 5 μmol $L^{-1}$) during ammonium oxidation. The main sources of ammonium in the Chesapeake Bay include remineralization of organic matter in the oxygenated water column and sediments (Kemp et al., 1990) and atmospheric deposition (Larsen et al., 2001). Onset of ammonium oxidation is viable at $NH_4^+$ concentration < 100 nmol $L^{-1}$ by the natural ammonia oxidizing community (Horak et al., 2013). Thus, $N_2O$ production from ammonium oxidation might be stimulated under low oxygen conditions by influx of ammonium near the oxic-anoxic interface, which deserves future research efforts.

Moreover, the relatively shallow oxic-anoxic interface means that $N_2O$ produced in the water column could be easily emitted to the atmosphere. In summertime (June to August), the typical depth of the oxic-anoxic interface is 10 – 15 m in the Chesapeake Bay (Taft et al., 1980; Kemp et al., 1992; Lee et al., 2015a). When storm events, boat traffic and surface cooling disturb the water column stratification, intermittent release of $N_2O$ to the atmosphere could occur.



## 4 Conclusion and outlook

The Chesapeake Bay is a potential $N_2O$ source via denitrification when $NO_3^-$ and $NO_2^-$ are present in low oxygen waters. Nitrogen (absolute and relative concentrations of $NO_3^-$ and $NO_2^-$) and oxygen availabilities control $N_2O$ production in the water column of Chesapeake Bay. Therefore the seasonal variation of nitrogen and oxygen availabilities (Lee et al.,

2015a) drive the seasonal variation in denitrifying community size, as shown by *nirS* gene abundance, and associated potential $N_2O$ production rates. The rate and occurrence of $N_2O$ production vary greatly between seasons; thus the annual rate of $N_2O$ production and consumption by the Bay and other estuarine systems is very difficult to estimate. The inhibition of $N_2O$ production by oxygen highlights the positive outcomes of re-oxygenation of the Chesapeake Bay. When elevated primary production in the surface layer is fueled by nitrogen input, aerobic remineralization at depth consumes oxygen

rapidly. In summertime, water column stratification restricts influx of oxygen to depth, creating seasonal anoxia/hypoxia in the Bay. The documented eutrophication and expansion of anoxia/hypoxia in the Chesapeake Bay in the late 20[th] century attracted public attention because of increasing mortality of organisms with high commercial and recreational value (Cooper and Brush, 1993). Moreover, expansion of the volume of low oxygen waters will result in more "hot spots" for $N_2O$ production. The key factor of mitigating anoxia is to control the nitrogen input to the bay (Hagy et al., 2004; Zhou et al.,

2014). This can be achieved by collaborative efforts of effective fertilizer application, sewage treatment, and natural nitrogen removal by microbial denitrification/anammox and plant uptake. Reducing the nitrogen input into estuaries such as the Chesapeake Bay will help mitigate $N_2O$ efflux: In the short-term, nitrogen sources ($NH_4^+$, $NO_2^-$ and $NO_3^-$) for $N_2O$ production will be decreased. In the long run, eutrophication will be alleviated, which will re-oxygenate the water column, and inhibit $N_2O$ production.





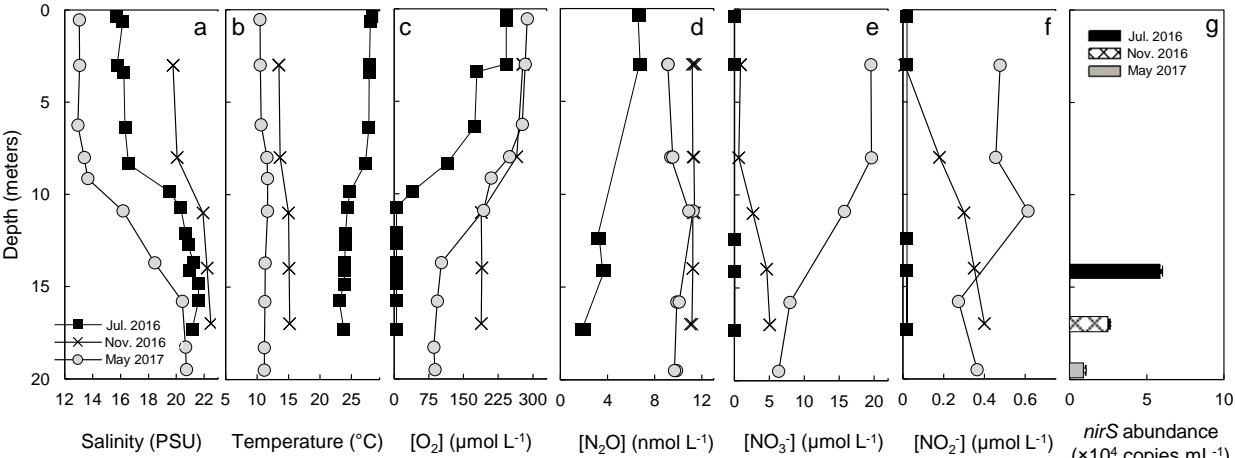

**Figure 1: Depth profiles on three sampling dates, July 19, 2016 (black square), November 17, 2016 (cross), May 3, 2017 (grey circle) of a) salinity, b) temperature, c) oxygen, d) nitrous oxide, e) nitrate, f) nitrite. Analysis of *nirS* gene abundance (g) was only conducted at one depth, at which incubations were also performed, during each trip.**

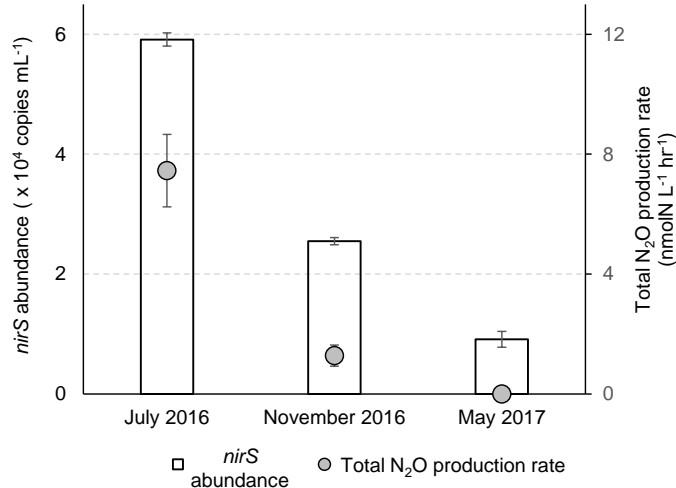

**Figure 2: Abundances of *nirS* gene and total N$_2$O production rates (from nitrate plus nitrite reduction) at three sampling times. The *nirS* gene abundances were analyzed at 14, 17 and 19 m during July 2016, November 2016 and May 2017, respectively. The total N$_2$O production rates were measured in the control experiment (helium-flushed anoxic incubation) at 12, 17 and 19 m during July 2016, November 2016 and May 2017, respectively.**



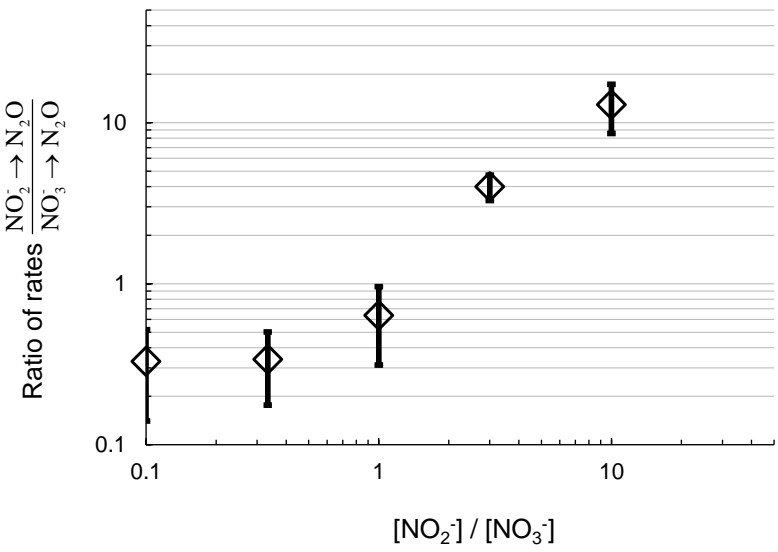

**Figure 3: Ratio of rates of N₂O production from NO₂⁻ reduction and NO₃⁻ reduction plotted with the respective ratio of NO₂⁻ to NO₃⁻ concentration in the DIN manipulation experiment from July 2016 sampling. Log scale on both axes is for clarity at the low values.**

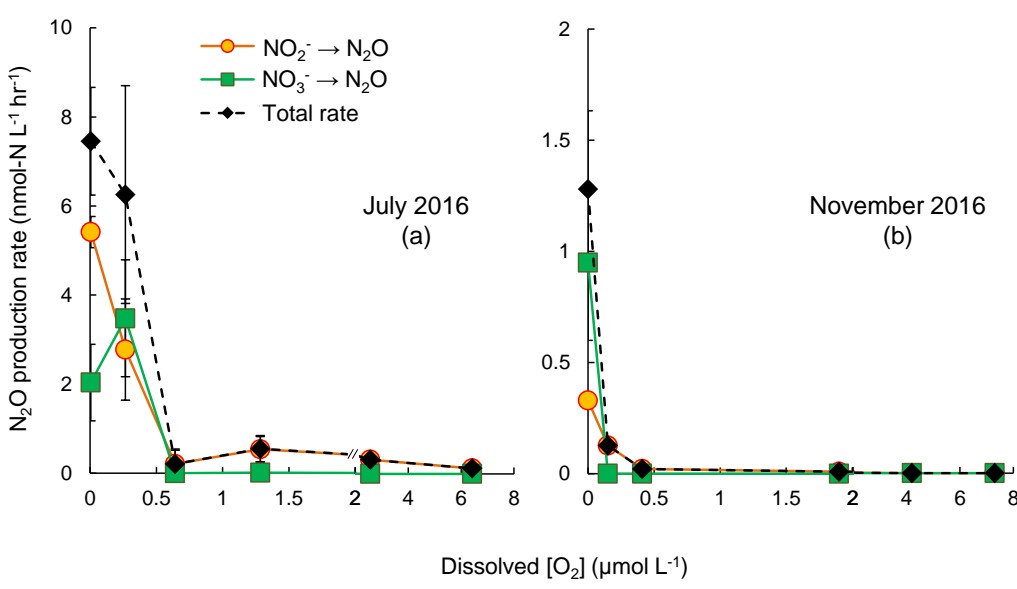

**Figure 4: Rates of N₂O production from NO₂⁻ reduction (orange circles), NO₃⁻ reduction (green squares) and combined NO₂⁻ and NO₃⁻ reduction (black diamonds) under increasing oxygen concentrations in July 2016 (a) and November 2016 (b). The standard deviation of rates in most of the samples were small so that error bars are not visible. Note the scale break at 2 μmol L⁻¹ O₂ on x-axis.**





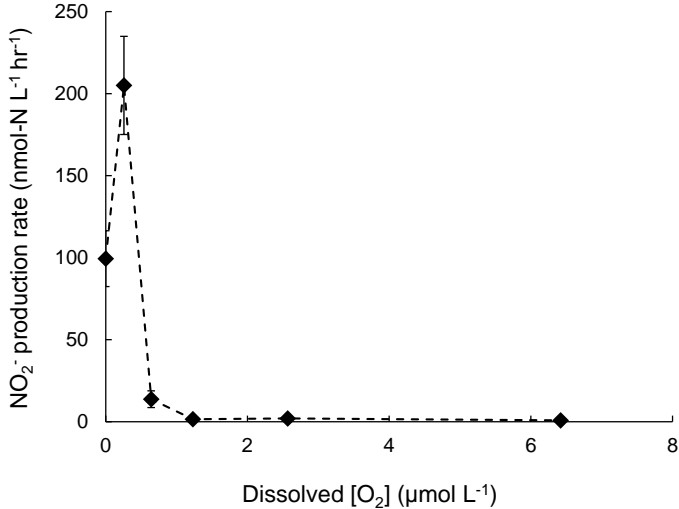

**Figure 5: Rates of NO₂⁻ production from NO₃⁻ reduction under increasing oxygen concentrations. Error bar indicates the standard**
5  **deviation of rates from linear regression of three time points (n=7).**





| Experiment | Experiment ID | $^{15}NO_2^-$ (µM) | $^{15}NO_3^-$ (µM) | $^{14}NO_2^-$ (µM) | $^{14}NO_3^-$ (µM) | $NO_2^-:NO_3^-$ | $^{15}N$ fraction label (species) | $O_2$ (µM) |
|---|---|---|---|---|---|---|---|---|
| **Control** | 1-A | 5 | | | 5 | 1:1 | 0.99 ($NO_2^-$) | 0 |
| **(July 2016)** | 1-B | | 5 | 5 | | 1:1 | 0.99 ($NO_3^-$) | 0 |
| **Nitrogen** | 2-A | 0.2 | | 1 | 10 | 1.2 : 10 | 0.16 ($NO_2^-$) | 0 |
| **manipulation** | 2-B | | 0.2 | 1 | 10 | 1 : 10.2 | 0.016 ($NO_3^-$) | 0 |
| **(July 2016)** | 2-C | 0.2 | | 1 | 3 | 1.2 : 3 | 0.16 ($NO_2^-$) | 0 |
| | 2-D | | 0.2 | 1 | 3 | 1: 3.2 | 0.06 ($NO_3^-$) | 0 |
| | 2-E | 0.2 | | 3 | 1 | 3.2 : 1 | 0.06 ($NO_2^-$) | 0 |
| | 2-F | | 0.2 | 3 | 1 | 3 : 1.2 | 0.16 ($NO_3^-$) | 0 |
| | 2-G | 0.2 | | 10 | 1 | 10.2 : 1 | 0.016 ($NO_2^-$) | 0 |
| | 2-H | | 0.2 | 10 | 1 | 10 : 1.2 | 0.16 ($NO_3^-$) | 0 |
| **Oxygen** | 3-A | 5 | | | 5 | 1:1 | 0.99 ($NO_2^-$) | 0.3 |
| **manipulation** | 3-B | | 5 | 5 | | 1:1 | 0.99 ($NO_3^-$) | 0.3 |
| **(July 2016)** | 3-C | 5 | | | 5 | 1:1 | 0.99 ($NO_2^-$) | 0.6 |
| | 3-D | | 5 | 5 | | 1:1 | 0.99 ($NO_3^-$) | 0.6 |
| | 3-E | 5 | | | 5 | 1:1 | 0.99 ($NO_2^-$) | 1.3 |
| | 3-F | | 5 | 5 | | 1:1 | 0.99 ($NO_3^-$) | 1.3 |
| | 3-G | 5 | | | 5 | 1:1 | 0.99 ($NO_2^-$) | 2.6 |
| | 3-H | | 5 | 5 | | 1:1 | 0.99 ($NO_3^-$) | 2.6 |
| | 3-I | 5 | | | 5 | 1:1 | 0.99 ($NO_2^-$) | 6.4 |
| | 3-J | | 5 | 5 | | 1:1 | 0.99 ($NO_3^-$) | 6.4 |
| **Control** | 4-A | 5 | | 0.4 | 10 | 0.54:1 | 0.93 ($NO_2^-$) | 0 |
| **(November 2016)** | 4-B | | 5 | 5.4 | 5 | 0.54:1 | 0.50 ($NO_3^-$) | 0 |
| **Oxygen** | 5-A | 5 | | 0.4 | 10 | 0.54:1 | 0.93 ($NO_2^-$) | 0.2 |
| **manipulation** | 5-B | | 5 | 5.4 | 5 | 0.54:1 | 0.50 ($NO_3^-$) | 0.2 |
| **(November 2016)** | 5-C | 5 | | 0.4 | 10 | 0.54:1 | 0.93 ($NO_2^-$) | 0.4 |
| | 5-D | | 5 | 5.4 | 5 | 0.54:1 | 0.50 ($NO_3^-$) | 0.4 |
| | 5-E | 5 | | 0.4 | 10 | 0.54:1 | 0.93 ($NO_2^-$) | 1.9 |
| | 5-F | | 5 | 5.4 | 5 | 0.54:1 | 0.50 ($NO_3^-$) | 1.9 |
| | 5-G | 5 | | 0.4 | 10 | 0.54:1 | 0.93 ($NO_2^-$) | 4.2 |
| | 5-H | | 5 | 5.4 | 5 | 0.54:1 | 0.50 ($NO_3^-$) | 4.2 |
| | 5-I | 5 | | 0.4 | 10 | 0.54:1 | 0.93 ($NO_2^-$) | 7.3 |
| | 5-J | | 5 | 5.4 | 5 | 0.54:1 | 0.50 ($NO_3^-$) | 7.3 |

Table 1: Parameters for control, nitrogen manipulation and oxygen manipulation incubation experiments in July 2016 and November 2016 sampling. The unit "µmol L$^{-1}$" is represented by "µM". Shaded columns highlight the concentrations for $^{15}N$ tracers. In situ nitrate and nitrite concentrations in July 2016 were < 0.02 µmol L$^{-1}$, and in November 2016 the concentrations were 5.0 and 0.4 µmol L$^{-1}$, repectively.





## 5 Funding Sources and Acknowledgements

This work is supported by the following funding sources: The PEI Grand Challenges – Control of Microbial Nitrous Oxide Production in Coastal Waters to B.B.W.. National Science Foundation (OCE 1427019) to J.C.C.. German Academic Exchange Service Postdoctoral Researchers International Mobility Experience fellowship to C.F. The authors would like to thank Michael Owens at Horn Point Laboratory for his assistance with field research equipment. We thank Sergey Oleynik for technical assistance during laboratory analysis.

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
