# Peer review of "Nitrogen and oxygen availabilities control water column nitrous oxide"

_Biogeosciences, 2018_

## Referee Comment (RC1) · Anonymous Referee #1 · 9 Apr 2018

The manuscript by Qixing Ji and colleagues investigates the controls on nitrous oxide concentrations in Chesapeake Bay. It is a topic that the authors are very familiar with and this expertise is reflected in the experimental design investigating the effect of nutrients and oxygen on nitrous oxide. The datasets are useful and should be published, but I recommend restructuring the manuscript to focus on the strengths of the work and dealing with the issues raised.

Major comments As can happen with studies which conduct repeated experiments at different time intervals with varying measurements, it is difficult at times to track all the activity. I believe there is a discrepancy between the Methods and Results, as Figure

2 shows N2O production rates for July, November, and May, yet in the Methods you state that the experiments were only conducted in July. Instances like this make it very difficult for the reader to follow.

I had a hard time relating the experiments with the estuarine environment. If you want to measure nitrous oxide emissions from Chesapeake Bay, the work conducted in this study is not what needs to be conducted. You would need high resolution surveys of key hydrographic parameters coupled with nitrous oxide measurements, either continuously or at targeted times. I recommend the authors focus more on the experiments as a means to better understand the controls on nitrous oxide production and use Chesapeake Bay as the contextual background, rather than attempting to explain nitrous oxide dynamics in this estuary.

Two examples of the mis-match between the datasets reported and the contextual description provided in the Introduction are (1) The abstract talks about intense N2O efflux from estuaries, but the results show nitrous oxide concentrations close to air-saturation values. (2) The title mentions seasonal anoxia, but this is not shown in the results.

I am not sure if the May 2017 dataset is necessary in Figure 1. It is helpful in Figure 2 only.

You need to include a description of how you calculate N2O production in the Methods section

The N2O profiles puzzle me in the context of the other results. It looks like July 2016 is the only profile which has significant differences with depth, decreasing in concentration between 3 and 13 m. However, this time point is associated with the highest rates of N2O production (Figure 2). Therefore, N2O consumption is very important yet hardly mentioned in the manuscript. In context of your comment that estuaries are emitting large quantities of N2O, the consumption processes deserve more attention.

You should explain to the reader why you focused on the nirS gene and not other relevant genes

Minor comments

Page 1 Line 16 Change reducing to decreasing

Page 2 Line 11 Agriculture such as paddy fields?

Page 4 Oxidized nitrogen. I think this refers to NO3 and NO2. I recommend you write NO3 and NO2 if this is correct, as it avoids NO or NOx. Also on Page 7, Line 18

Page 3 Line 21 It would help orientate the reader if you provide a short explanation for why you chose these 3 depths. For example, why did you only sample below the oxycline in anoxic waters and out of curiosity, why did you not compare anoxic and oxic?

Page 4 Line 14 Why do you inject N2O to detect N2O production? Is it an issue of detection limit? Or were you looking for N2O consumption?

Page 4 Line 14 If you inject 1.2 nmol, how do you get 20 nmol L-1? Presumably you verified these target concentrations on a few bottles and you should state the final concentrations achieved.

Page 5 Line 1. Again, presumably you checked the final concentrations of oxygen against your target concentrations. I suspect you also did the air-equilibration at a single temperature which you should state.

Page 5, Line 19 The alternative to injecting N2O standards into crimp-sealed vials is to air-equilibrate at controlled temperatures. This might be easier?

Page 7, Line 13 Where is the intense efflux that you mentioned in the abstract?

Page 7, Line 13 If you are going to talk about saturation, you have to provide the saturation value for each of the three sampling occasions

Page 8, Line 10 Are you surprised that there is a difference between a study in 1978 and a single experiment conducted 40 years later?

---

## Referee Comment (RC2) · Anonymous Referee #2 · 11 Apr 2018

General Comments:

The authors have presented a well-designed and well-executed experiment on N2O production in Chesapeake Bay. While the experiments and data are worthy of publication, the manuscript itself needs major revisions before it is accepted for publication. In reading the manuscript the introduction and methods were written clearly and concisely, but the results and discussion needs a substantial reworking. The paragraphs jumped from one topic to the next and often I found the subtitled headings were not appropriate for the range of topics covered in the text below. In the abstract the authors suggest two potential impacts of what reducing N to the bay will have on N2O produc-

tion. I think the paper is missing a more detailed explanation of how the experiments done lead to the conclusions they make, and they should expand on the specific details of these impacts that they suggest. I also think the authors could elaborate on the limitations of their study and what could be done in the future.

Minor Comments:

Abstract:

Pg1 line15: "N2O production was positively correlated with the ratio of nitrate to nitrite concentrations." Please clarify if this was in situ concentrations or the nitrate and nitrite added to the incubation.

Pg1 line18-19: What do you mean by nitrogen deficiency? What lengths of time are your referring to when you write "short term" and "in the long-run", can you approximate the time scale?

Methods:

Pg4 line15: Why do you add 20 nM N2O before the incubation? If it is so you can have enough N2O to measure the N2O production, could you add it after the incubation is killed? Can you clarify here in the text? Do you think adding it before, could affect the production of N2O?

Pg4 line17: What was the total concentration of nitrite+nitrate of tracer added?

Pg5 line8: How does 5uM of tracer compare to the in situ amounts of nitrite and nitrate.

Section 3.2 Active N2O production by denitrification

Pg8 line3: In the title you suggest all N2O is from denitrification, it would be good discuss why it is not from nitrification, if you are using that as the section title.

Pg8 line16: I think it's good you say here they are potential rates of N2O production, because you are removing all the oxygen, even if that's not the in situ value. I am

curious why there are no potential rates in May. Can you add some discussion on why removing the oxygen does not stimulate N2O production in May versus November?

Pg9 lines1-13: This section seems disjointed from the paragraph above and the title of the section. This paragraph refers more to removal of fixed nitrogen from the bay rather than N2O production.

Section 3.3 N2O production pathways regulated by availability of nitrogen substrate

Pg9 line16: I was confused by this term "NO2- (NO3-), is this synonymous with "NO2- or NO3- "? If so, I would suggest changing it to the latter.

Pg10 lines1-21, pg 11 lines1-8: This could use a different subheading? Could you reformat the equations to be easier to read? Also in general this section could use a little more description, as is, it is a little hard to follow. I would start off the section with the punch line (on pg 11) and then describe why the calculations back up that statement.

Section 3.4 Oxygen inhibits N2O production by denitrification

Title: I suggest changing the title to something more detailed.

Pg11 lines11-19: These sentences seem to belong to the other section about N influx to the bay. They could be moved or deleted during restructuring of discussion sections because it is somewhat repetitive from before.

Pg12 lines9-15: The result that there is a different oxygen tolerance for nitrite vs. nitrate reduction is really interesting. I think here is where you should expand on more why you get this result. I know you touch on it more in a later section, but it might be best to put it all together (maybe in it's own section). Could it be from nitrifier denitrification? How would nitrite oxidation affect the production rate? At low levels of oxygen I would suspect there would be some nitrite oxidation to nitrate.

Pg12 lines16-19: This paragraph seems out of place? Again, would you suspect some

nitrite oxidation in these nitrate reduction to nitrite measurements? How would that affect results?

Pg13 lines3-4: This line should go with the section on differences of nitrite-stimulated production vs. nitrate-stimulated N2O production.

Pg13 lines10-18: This paragraph is out of place.

Pg13 lines19-22: Should this be in the conclusion and outlook?

Section 4 Conclusion and outlook

Pg14 lines14-19: I would reorder this section and put these lines first.

Conclusion and outlook section as a whole: Could you add something connecting the two pathways of management (nitrogen influx vs. oxygenation)? Also, could you quantify how each change inhibits N2O production?

---

## Referee Comment (RC3) · Anonymous Referee #3 · 13 Apr 2018

Summary

The authors present an examination of N2O dynamics studied in the Chesapeake Bay during three samplings. Since the Chesapeake Bay exhibits a strong seasonal shift in water column redox state, the study focused on trying to link these shifts with N2O production mechanisms at and below the oxic/anoxic interface. The authors bring a range of chemical, molecular and isotopic tools to bear on these dynamics, with an emphasis on elucidating N2O producing processes occurring in the bottom waters and the primary controls on them. This contribution is timely – as coastal and estuarine systems are dynamic and generally understudied with respect to their place in the

global N2O budget. Overall the data appear to be of high quality. The manuscript is generally well written, though parts could benefit from some reorganization. I have some questions about the data interpretation as outlined below. Overall I think this work is worthy of publication, but that the manuscript could be improved through some more careful consideration of clarifying some sections.

Major Comments

Pg 9 Ln 15: I appreciate the use of targeted assays for NO2- and NO3- reduction, though it is unclear whether this was designed to constrain/target nitrifier denitrification specifically or explicit nitrite reducing denitrifiers (???). Clearly denitrifying organisms also use NO2- in their electron transport chain. In Section 3.3 the authors attempt to tackle this – and I appreciate the argument that they are making about NO2- transport across the membrane – but I feel that this section is confusing as written. Using calculations laid out here, and making a few key assumptions, the authors conclude that since the level of 15N label in the N2O pool is much higher than if there had been full exchange, then exchange between the cellular and ambient NO2- is minimal. I acknowledge that this is a difficult aspect of N cycling to track, but I am not overly convinced that they have proven that this type of exchange is 'minimal.' Their calculation demonstrates that high levels of exchange are not occurring, but whether modest levels might be influencing the results is unclear. Perhaps this argument could be streamlined and clarified.

Additionally, perhaps the introduction needs some sort of clearer description of the types of metabolisms being targeted by the study (complete denitrifiers, nitrite denitrifiers, nitrifier denitrification). These classifications of microbes and processes are confusing even to those who regularly study N cycling.

Minor Comments

Pg 1 Ln15: I believe nitrate and nitrite are reversed here (and many other times throughout – leading to some frustration/confusion).

Pg 1 Ln17: Since the field data demonstrate that there is no net flux to the atmosphere – it seems odd to emphasize N2O efflux here.

Pg 3 Ln 19: Please clarify whether a headspace was left in the incubation bottle or not.

Pg 4 Ln 1: course not courses

Pg 5 Ln 3: Please indicate whether oxygen concentrations were measured or calculated?

Pg 6 Ln 1: Why were no other functional gene assays performed? I think this is justified later, but given that nirS only reflects denitrification – its linkage with N2O from this pathway is clear, yet reveals little about the dynamics of the other pathways investigated as I understand. Why not include nirK? Or norB?

Pg 7 Ln 19: I believe nitrate and nitrite are reversed here again.

Pg 7 Ln 25: "positively correlates" – yes, but this is difficult to defend statistically with n=3.

Pg 13 Ln 2: I would suggest "microbial groups" instead of microbial communities (which may imply the 'greater community' – not just N cycling organisms ?).

Pg 14 Ln 5: It seems that if nitrifier denitrification and ammonia oxidation are implicated in N2O production as discussed – then the nitrifier community dynamics would also play an important role and should be acknowledged?

---

## Referee Comment (RC4) · Anonymous Referee #4 · 16 Apr 2018

The paper reports an experimental study of rates and pathways of nitrous oxide production in Chesapeake Bay waters. Water was sampled on three occasions (spring, summer, autumn) and incubated with N-15 labelled nitrate or nitrite under anoxic conditions. Additional incubations were made with oxygen added back to investigate the oxygen sensitivity of the processes. Based on the results, the authors draw conclusions about the controls of N2O emissions from the Bay.

The paper addresses an interesting subject and the experimental work is of good quality. However, the results do not provide strong support for the conclusions because the experimental conditions do not sufficiently reflect the environmental conditions in

the Bay. Also, although there are few previous experimental studies from comparable environments, the paper largely neglects the large number of previous studies on N2O dynamics in estuaries, although these do provide some insight to the controls of N2O emission. Importantly, the literature points to nitrification (ammonium oxidation) as a major N2O source in estuaries whereas the present study only investigates N2O production through denitrification. Without data on the rates and controls of N2O production by ammonia oxidation (i.e. experiments with N-15 labelled ammonia at different oxygen concentrations), no conclusions can be drawn about the controls of N2O emissions from Chesapeake Bay.

Based on the mismatch between the experiments and the conclusions, I recommend that the paper be rewritten to focus on what the experiments can actually tell us, i.e. how N2O production during denitrification is affected by oxygen, and how production from nitrate and nitrite seem to function independently, which is novel and interesting. I also warn against trying to translate the results into understanding denitrification as a N2O source in the Bay as a whole, and the role of anoxia in this, because denitrification is an interface process there, and the anoxic water body might serve as net sink for N2O, drawing it down from the overlying oxycline. Here, fine scale profiling of N2O across the interface might be more informative than the experimental approach.

Specific comments 3, 3: Why pilot? – to me this indicates preliminary results

7, 10: The detection limit for H2S by smell is $\sim$10 $\mu$M. It seems strange if no sulphide was present at all, if all more favourable oxidants were depleted.

9, 5: I would leave out this back-of-the-envelope estimate of denitrification. It does not add new, robust insight to N loss in C. B.

9, 23 and onwards: This is an important finding, which requires elaboration. I suggest calculating the direct contribution from nitrate to N2O for all the different combinations of nitrate and nitrite concentrations instead of just one example. If rates are assumed to be constant during the incubation, a simple model can describe the concomitant

production and consumption of nitrite and hence how N-15 should accumulate in the extracellular nitrite pool if the intermediate nitrite were exchanging freely.

10, 9: I don't understand this formula. How can the amount of 15N-nitrite produced depend on either the total nitrate and the total nitrite concentration? Shouldn't it simply be: Rate of NO2- production from NO3- $\times$ incubation time $\times$ initial fraction labelled of NO3-?

12, 1: don't understand this. Oxygenation will just shift the zone of denitrification and N2O consumption to greater depth (until it reaches the sediment). It won't necessarily inhibit it.

―――――――――――――――――――

---

## Author Comment (AC1) · 16 May 2018

[Referee] The manuscript by Qixing Ji and colleagues investigates the controls on nitrous oxide concentrations in Chesapeake Bay. It is a topic that the authors are very familiar with and this expertise is reflected in the experimental design investigating the effect of nutrients and oxygen on nitrous oxide. The datasets are useful and should be published, but I recommend restructuring the manuscript to focus on the strengths of the work and dealing with the issues raised.

Major comments [Referee] As can happen with studies which conduct repeated experiments at different time intervals with varying measurements, it is difficult at times to

track all the activity. I believe there is a discrepancy between the Methods and Results, as Figure 2 shows N2O production rates for July, November, and May, yet in the Methods you state that the experiments were only conducted in July. Instances like this make it very difficult for the reader to follow.

[Response] The N2O production rates shown in figure 2 were measured in control incubations, which were performed on all three sampling dates. In control incubations, samples received 5 $\mu$mol L-1 15N-nitrate or 15N-nitrite, respectively; and oxygen was removed from samples by helium flushing. The DIN manipulation experiments were only conducted in July. We explained this in page 5 and revised table 1 to minimize confusion.

[Referee] I had a hard time relating the experiments with the estuarine environment. If you want to measure nitrous oxide emissions from Chesapeake Bay, the work conducted in this study is not what needs to be conducted. You would need high resolution surveys of key hydrographic parameters coupled with nitrous oxide measurements, either continuously or at targeted times. I recommend the authors focus more on the experiments as a means to better understand the controls on nitrous oxide production and use Chesapeake Bay as the contextual background, rather than attempting to explain nitrous oxide dynamics in this estuary.

[Response] We agree with the reviewer's suggestion, and this focus on control is exactly what we intended, with the title stating explicitly that this manuscript is about examination of the control of N2O production, rather than emissions, from the Chesapeake Bay. We mention briefly that water column N2O the may be emitted due to disruption of water column stratification (page 13), which is a motivation for the research and points to further research directions. We'll revise accordingly in the next version.

[Referee] Two examples of the mis-match between the datasets reported and the contextual description provided in the Introduction are (1) The abstract talks about intense

N2O efflux from estuaries, but the results show nitrous oxide concentrations close to air-saturation values. (2) The title mentions seasonal anoxia, but this is not shown in the results.

[Response] For (1), this is a more of a "general" statement that estuaries can be sites of intense N2O efflux, and previous studies have shown that the Chesapeake Bay is a N2O source. This context motivated our research. The measurements and experiments showed that despite the fact that the water column is close to N2O saturation level, N2O production can occur. We further explain in page 11. For (2) The oxygen profiles obtained from this study (and many others) show that water column anoxia occurs in summer. In spring and autumn, the water column is oxygenated. These results demonstrate the water column anoxia is a seasonal event, as clearly shown in the two papers by Lee et al., which are cited in the manuscript. A revised abstract will be presented in the next version to minimize confusion to the readers.

[Referee] I am not sure if the May 2017 dataset is necessary in Figure 1. It is helpful in Figure 2 only.

[Response] The May dataset illustrates seasonal variation in the usual parameters, and thus supports the seasonal nature of N2O production and consumption. In May 2017, low oxygen condition was forming, along with higher availability of nitrate. These conditions are becoming favorable for denitrification in the water column.

[Referee] You need to include a description of how you calculate N2O production in the Methods section.

[Response] The rate calculation is presented in section 2.4, equation 1

[Referee] The N2O profiles puzzle me in the context of the other results. It looks like July 2016 is the only profile which has significant differences with depth, decreasing in concentration between 3 and 13 m. However, this time point is associated with the highest rates of N2O production (Figure 2). Therefore, N2O consumption is very

important yet hardly mentioned in the manuscript. In context of your comment that estuaries are emitting large quantities of N2O, the consumption processes deserve more attention.

[Response] N2O consumption is indeed important; however it was not quantified nor addressed directly in the current manuscript. The work here focuses on the controls of N2O production via denitrification. One of the future research direction is to examine the N2O reduction during anoxic events. The fact that significant N2O production was detected in the month when N2O concentration was lowest implies that N2O consumption was also occurring and probably minimizing efflux to the atmosphere.

[Referee] You should explain to the reader why you focused on the nirS gene and not other relevant genes

[Response] Good point. The nirS gene encodes the genetic material for nitrite reductase, the enzyme responsible for nitrite reduction to nitric oxide. NirS is often used as a proxy for the abundance and diversity of denitrifying bacteria (which was our application here) and is the gene in the denitrification sequence that is most reliably associated with a complete denitrification pathway (Graf et al. 2014).

Minor comments [Referee] Page 1 Line 16 Change reducing to decreasing

[Response] Corrected.

[Referee] Page 2 Line 11 Agriculture such as paddy fields?

[Response] Is the reviewer referring to "agricultural land" in line 13? If so, paddy fields are not common in the Chesapeake Bay watershed. Agriculture is the main land use in this region and major agricultural activities in the area include livestock farming, greenhouse and nursery products (flowers, ornamental shrubs, and young fruit trees), corn and soybeans, all of which depend heavily on industrial fertilizers.

[Referee] Page 4 Oxidized nitrogen. I think this refers to NO3 and NO2. I recommend you write NO3 and NO2 if this is correct, as it avoids NO or NOx. Also on Page 7, Line

[Response] Corrected.

[Referee] Page 3 Line 21 It would help orientate the reader if you provide a short explanation for why you chose these 3 depths. For example, why did you only sample below the oxycline in anoxic waters and out of curiosity, why did you not compare anoxic and oxic?

[Response] As stated in the introduction, the anoxic events in the Chesapeake Bay is of great environmental and economic concerns. One of the motivations of this work is to examine the control of N2O production under anoxic condition. We sampled oxygenated condition in May and November because anoxic conditions were not detected at the depths where they would have been expected during the seasonal anoxia, thus the oxic conditions were sampled for comparison and, not surprisingly, had lower rates of denitrification. It is likely that nitrification is important during oxygenated conditions. Future work more focus more on the oxic waters.

[Referee] Page 4 Line 14 Why do you inject N2O to detect N2O production? Is it an issue of detection limit? Or were you looking for N2O consumption?

[Response] It is an issue of detection limit of mass spectrometer, which requires > 2 nmol of nitrogen.

[Referee] Page 4 Line 14 If you inject 1.2 nmol, how do you get 20 nmol L-1? Presumably you verified these target concentrations on a few bottles and you should state the final concentrations achieved.

[Response] N2O concentration in the incubation bottles was estimated as follows: The incubation bottle is 60 ml in volume. Considering only 3 ml of headspace in the bottle, 90 – 95 % will dissolve in water phase (57 ml, 0.057 L) under experimentally relevant conditions. Therefore, 1.2 nmol $\times$ 0.9 / 0.057 L $\approx$ 20.0 nmol L-1. N2O concentration was directly measured in the time course samples on the mass spec, so no assump-
tions were necessary in the actual rate calculations.

[Referee] Page 5 Line 1. Again, presumably you checked the final concentrations of oxygen against your target concentrations. I suspect you also did the air-equilibration at a single temperature which you should state.

[Response] Oxygen saturated site water was made by vigorously shaking an open-capped bottle with water collected at depth. Concentration was calculated using formula from Gordon and Garcia, 1992, at measured temperature (< 0.5 degC difference of in situ) and salinity. Oxygen concentrations were not measured directly in these experiments. We have, however, used an optical sensor to measure concentrations directly in the same kinds of bottles in similar experiments and the agreement between estimated target concentration and measured is excellent. Thus it is not necessary to measure every bottle every time.

[Referee] Page 5, Line 19 The alternative to injecting $N_2O$ standards into crimp-sealed vials is to air-equilibrate at controlled temperatures. This might be easier?

[Response] The measurement of $N_2O$ is using purge-and-trap technique, which analyze all of $N_2O$ contained in the vials by a mass spectrometer. Thus, the total amount of $N_2O$ is important, even the $N_2O$ standards may not be in equilibrium between the water and headspace.

[Referee] Page 7, Line 13 Where is the intense efflux that you mentioned in the abstract?

[Response] As explained earlier, this refers to a previous study documenting the Chesapeake Bay as a $N_2O$ source. Conditions in estuaries are highly variable, in both time and space, which is one of the motivations for investigating the control mechanisms on $N_2O$ production. Failure to detect efflux at this time does not mean that intense efflux does not occur at this site at other times, or in other parts of the Bay.

[Referee] Page 7, Line 13 If you are going to talk about saturation, you have to provide

the saturation value for each of the three sampling occasions.

[Response] The surface N2O saturation values in July, November and May are: 6.6, 10.4 and 12.0 nmol L-1, respectively. These values have been included in the revised text.

---

## Author Comment (AC2) · 16 May 2018

General Comments: [Referee] The authors have presented a well-designed and well-executed experiment on N2O production in Chesapeake Bay. While the experiments and data are worthy of publication, the manuscript itself needs major revisions before it is accepted for publication. In reading the manuscript the introduction and methods were written clearly and concisely, but the results and discussion needs a substantial reworking. The paragraphs jumped from one topic to the next and often I found the subtitled headings were not appropriate for the range of topics covered in the text below. In the abstract the authors suggest two potential impacts of what reducing N

to the bay will have on N2O production. I think the paper is missing a more detailed explanation of how the experiments done lead to the conclusions they make, and they should expand on the specific details of these impacts that they suggest. I also think the authors could elaborate on the limitations of their study and what could be done in the future.

[Response] The reviewer's major criticism concerns the structure of results and discussion, which we will explain later. The main objective of this work is to examine the control of water column N2O production via denitrification in the Bay. The experiments were designed to suit our objective, by increasing the availability of oxygen and dissolved inorganic nitrogen through manipulation, as well as measuring N2O production in different seasons. In doing so we concluded that increasing DIN availability and decreasing oxygen availability could stimulate denitrification during summertime anoxia. In fall and spring the N2O production rates via denitrification were low, probably due to lack of anoxic conditions and concomitant low abundance of denitrifiers.

The major limitation of our study, like other tracer incubation studies in aquatic environments, is that the N2O production rates measured here should be treated as potential rates because the experimental conditions were not 100% identical to in situ environments. Despite these artifacts, the conclusions that DIN and oxygen availabilities control N2O production during anoxic events are robust. The current work could be complemented by these future research directions, (1) measurement of nitrification rates in oxic waters and associated N2O production and nitrification genes; (2) measurement of N2O reduction rates within anoxic layer and characterization of N2O dynamics in the Chesapeake Bay; (3) Elucidation of intracellular nitrite exchange during nitrate reduction in natural waters; (4) Investigation of the physical, chemical and biological controls of N2O emission in the Chesapeake Bay. A revised "Conclusion and outlook" section will incorporate the above in the next version.

Minor Comments: Abstract: [Referee] Pg1 line15: "N2O production was positively correlated with the ratio of nitrate to nitrite concentrations." Please clarify if this was in

situ concentrations or the nitrate and nitrite added to the incubation.

[Response] This statement describes N2O production rates under manipulated nitrate and nitrite concentrations. All of rates reported in the manuscript were measured using 15N tracers, meaning the substrate concentrations (nitrate and nitrite) were higher than in situ levels. We revised the sentence in the new version.

[Referee] Pg1 line18-19: What do you mean by nitrogen deficiency? What lengths of time are your referring to when you write "short term" and "in the long-run", can you approximate the time scale?

[Response] The term "nitrogen deficiency" refers to nitrate and nitrite concentrations below detection. This occurs during 4 months of summer from June to September. We use "short-term" to describe a time scale of months, and "long-run" of years. We will revise the sentence in the next version to improve clarity.

Methods: [Referee] Pg4 line15: Why do you add 20 nM N2O before the incubation? If it is so you can have enough N2O to measure the N2O production, could you add it after the incubation is killed? Can you clarify here in the text? Do you think adding it before, could affect the production of N2O?

[Response] It is an issue of detection limit of mass spectrometer, which requires > 2 nmol of N. We agree that the added tracer concentration is slightly higher than in situ (6 – 12 nM). It is our intention to add the N2O before the samples were preserved so as to have a more similar condition to in situ. The effect of N2O concentration on the rate of N2O production should be investigated.

[Referee] Pg4 line17: What was the total concentration of nitrite+nitrate of tracer added?

[Response] Here is the description of DIN manipulation experiment, which was conducted in July 2016 only. The total concentrations of nitrate and nitrite were 5 $\mu$M each, and all the substrate concentrations of every experiments conducted are listed

in table 1.

[Referee] Pg5 line8: How does 5uM of tracer compare to the in situ amounts of nitrite and nitrate.

[Response] This was stated in page 7. In July 2016, in situ nitrate and nitrite concentrations were below detection (0.02 $\mu$M), and the experimental conditions are in no way representing the in situ. Hence we treated the rates as potential rates. In November 2016, in situ nitrate and nitrite concentrations at the depth of experiment (17 m) were 5.0 and 0.4 $\mu$M, respectively.

Section 3.2 Active N2O production by denitrification [Referee] Pg8 line3: In the title you suggest all N2O is from denitrification, it would be good discuss why it is not from nitrification, if you are using that as the section title.

[Response] Indeed. We only measured N2O production via denitrification. Unfortunately we did not measure N2O production via nitrification. We will change the title to "Active N2O production in the water column" to minimize confusion.

[Referee] Pg8 line16: I think it's good you say here they are potential rates of N2O production, because you are removing all the oxygen, even if that's not the in situ value. I am curious why there are no potential rates in May. Can you add some discussion on why removing the oxygen does not stimulate N2O production in May versus November?

[Response] We explained in the following paragraph. This is likely due to low denitrifier abundance (indicated by nirS gene abundance) in May, during which the anoxic condition had not yet developed. In addition, our anoxic incubation lasted around 2 hours; it is unlikely for denitrifiers to reactivate within such a short time scale. We'll add the above information in the next version.

[Referee] Pg9 lines1-13: This section seems disjointed from the paragraph above and the title of the section. This paragraph refers more to removal of fixed nitrogen from the bay rather than N2O production.

[Response] Indeed. We will remove this paragraph in the next version.

Section 3.3 N2O production pathways regulated by availability of nitrogen substrate

[Referee] Pg9 line16: I was confused by this term "NO2- (NO3-), is this synonymous with "NO2- or NO3- "? If so, I would suggest changing it to the latter.

[Response] This is simply a combination of two sentences in one: ". . . increasing nitrite availability favors N2O production from nitrite reduction" and ". . . increasing nitrate availability favors N2O production from nitrate reduction." We change the sentence to "This suggests increasing NO2- or NO3- availability favors N2O production from the reduction of respective substrate."

[Referee] Pg10 lines1-21, pg 11 lines1-8: This could use a different subheading? Could you reformat the equations to be easier to read? Also in general this section could use a little more description, as is, it is a little hard to follow. I would start off the section with the punch line (on pg 11) and then describe why the calculations back up that statement.

[Response] We think another sub-heading is probably not necessary but we will reorganize the section in the next version. We'll add the conclusion "the exchange between intracellular and ambient nitrite during nitrate reduction to N2O is limited" in the beginning of the section. The calculation is to find the 15N fraction label of N2O when nitrite is fully exchanged, and we think it is rather straightforward. We'll use a different font and size to better illustrate the calculation in the next version.

Section 3.4 Oxygen inhibits N2O production by denitrification

[Referee] Title: I suggest changing the title to something more detailed.

[Response] The title states the major conclusion of this section and seems pretty specific.

[Referee] Pg11 lines11-19: These sentences seem to belong to the other section about

N influx to the bay. They could be moved or deleted during restructuring of discussion sections because it is somewhat repetitive from before.

[Response] Indeed. This paragraph will be organized in the "conclusion and outlook" section.

[Referee] Pg12 lines9-15: The result that there is a different oxygen tolerance for nitrite vs. nitrate reduction is really interesting. I think here is where you should expand on more why you get this result. I know you touch on it more in a later section, but it might be best to put it all together (maybe in it's own section). Could it be from nitrifier denitrification? How would nitrite oxidation affect the production rate? At low levels of oxygen I would suspect there would be some nitrite oxidation to nitrate.

[Response] It is possible that nitrifier denitrification is responsible for part of N2O production via nitrite reduction. This is because increasing oxygen availability will inhibit the activity of denitrifiers, but not nitrifiers. It is not possible to have a complete explanation with the data presented here. We'll add the above in the next version.

We did not measure nitrite oxidation rates in the samples but agree that it is likely nitrite oxidation was also occurring. We expect nitrite oxidation to have a negligible effect on N2O production rates, however, since N2O is not a product or intermediate in that reaction and expected rates of nitrite oxidation would not significantly affect the concentration of the substrates directly involved in N2O production. (Nitrite oxidations rates in nM/d would not affect the substrate concentrations, which were in the micromolar range.)

[Referee] Pg12 lines16-19: This paragraph seems out of place? Again, would you suspect some nitrite oxidation in these nitrate reduction to nitrite measurements? How would that affect results?

[Response] See above.

[Referee] Pg13 lines3-4: This line should go with the section on differences of nitritestimulated production vs. nitrate-stimulated N2O production. Pg13 lines10-18: This paragraph is out of place. Pg13 lines19-22: Should this be in the conclusion and outlook?

[Response] The section 3.4 will be re-organized in the next version to improve clarity and make tighter connections between the data and conclusions.

[Referee]Section 4 Conclusion and outlook Pg14 lines14-19: I would reorder this section and put these lines first. Conclusion and outlook section as a whole: Could you add something connecting the two pathways of management (nitrogen influx vs. oxygenation)? Also, could you quantify how each change inhibits N2O production?

[Response] Indeed. The current work will be complemented by these future research directions, (1) measurement of nitrification rates in oxic waters and associated N2O production and nitrification genes; (2) measurement of N2O reduction rates within anoxic layer and characterization of N2O dynamics in the Chesapeake Bay; (3) Elucidation of intracellular nitrite exchange during nitrate reduction in natural waters; (4) Investigation of the physical, chemical and biological controls of N2O emission in the Chesapeake Bay. A revised "Conclusion and outlook" section will incorporate the above in the next version.

---

## Author Comment (AC4) · 16 May 2018

[Referee] The paper reports an experimental study of rates and pathways of nitrous oxide production in Chesapeake Bay waters. Water was sampled on three occasions (spring, summer, autumn) and incubated with N-15 labelled nitrate or nitrite under anoxic conditions. Additional incubations were made with oxygen added back to investigate the oxygen sensitivity of the processes. Based on the results, the authors draw conclusions about the controls of N2O emissions from the Bay. The paper addresses an interesting subject and the experimental work is of good quality. However, the results do not provide strong support for the conclusions because the experimental

conditions do not sufficiently reflect the environmental conditions in the Bay.

Also, although there are few previous experimental studies from comparable environments, the paper largely neglects the large number of previous studies on N2O dynamics in estuaries, although these do provide some insight to the controls of N2O emission.

Importantly, the literature points to nitrification (ammonium oxidation) as a major N2O source in estuaries whereas the present study only investigates N2O production through denitrification. Without data on the rates and controls of N2O production by ammonia oxidation (i.e. experiments with N-15 labelled ammonia at different oxygen concentrations), no conclusions can be drawn about the controls of N2O emissions from Chesapeake Bay.

Based on the mismatch between the experiments and the conclusions, I recommend that the paper be rewritten to focus on what the experiments can actually tell us, i.e. how N2O production during denitrification is affected by oxygen, and how production from nitrate and nitrite seem to function independently, which is novel and interesting.

I also warn against trying to translate the results into understanding denitrification as a N2O source in the Bay as a whole, and the role of anoxia in this, because denitrification is an interface process there, and the anoxic water body might serve as net sink for N2O, drawing it down from the overlying oxycline. Here, fine scale profiling of N2O across the interface might be more informative than the experimental approach.

[Response] The reviewer's major criticism is that the manipulated experimental conditions led to the conclusion of the Chesapeake Bay being a N2O source, whereas the static concentration profile suggested the Bay as a N2O sink. We shall explain as follows:

(1) Our work of environmental control of N2O production is motivated by previous studies that identified the Chesapeake Bay as a N2O source (Elkins et al. 1978; McElroy

et al., 1978).

(2) Through incubation experiments, it is straightforward to draw the conclusion from the results: Adding nitrogen substrates and removing oxygen stimulate N2O production in summer and autumn. Thus the Bay is potentially a N2O source when pulses of nitrogen enters the water body that is experiencing summertime anoxia. Conditions in estuaries are highly variable, in both time and space, which is one of the motivations for investigating the control mechanisms on N2O production. Failure to detect efflux at this time does not mean that intense efflux does not occur at this site at other times, or in other parts of the Bay.

(3) Indeed there's a large body of literature reporting the variation of N2O fluxes in estuaries around the world. Many of them applied the traditional methodology: By quoting surface N2O supersaturation, wind speed, temperature, water turbulence, etc. and measure N2O fluxes, which is then related to water column oxygen and nitrogen availability. A small number of work applied nitrogen isotopic approach to study the mechanism of N2O production. The isotopic approach reveals the dynamic, potential N2O production that is masked by the static concentration profile. In addition, the isotopic approach allows to quantify the environmental controls of N2O production in the water column.

We disagree with the reviewer about the lack of similarity between experimental and actual conditions. Our incubation experiments were designed to study the effects of oxygen and nitrogen availability on N2O production, by changing experimental nitrogen ($1 - 10$ $\mu$M) and oxygen concentrations ($0 - 10$ $\mu$M) that occur regularly in the Chesapeake Bay (Lee et al., 2015).

As we have stated in the Introduction, nitrification is another important pathway for N2O production. The focus of the manuscript is N2O production under naturally occurring and laboratory anoxic condition. It is unlikely that nitrification could occur, and thus nitrification is beyond the scope of this manuscript. We thank the reviewer for pointing

out one possible future research direction: study the N2O production via nitrification and denitrification and their environmental controls with extended spatial and temporal coverage across the Chesapeake Bay.

The scope of this manuscript is to examine the control of N2O production by nitrogen and oxygen availability during denitrification under naturally occurring and laboratory anoxic condition. It is another future research direction that study the N2O dynamics by fine scale profiling of N2O across the interface so as to demonstrate whether the anoxic Chesapeake Bay serves as a net sink or source for N2O. We will revise accordingly in the next version.

Specific comments [Referee] 3, 3: Why pilot? – to me this indicates preliminary results

[Response] We chose "pilot" because we are the first group to study N2O production using 15N tracer at a single station in the Chesapeake Bay, and we examined the environmental control of denitrification pathway for N2O production. This is a small scale study and the results are important for a larger scale, more comprehensive study in the future.

[Referee] 7, 10: The detection limit for H2S by smell is $\sim 10 \mu$M. It seems strange if no sulphide was present at all, if all more favourable oxidants were depleted.

[Response] It would be more helpful if the reviewer list the reference. First, the odor threshold of H2S is $\sim$ 0.5 ppb [1], and Henry's Law constant of H2S is $\sim$ 0.1 mol/kg/bar [2]. Under atmospheric pressure with 0.5 ppb H2S, the equilibrium concentration of a H2S solution is $0.5 \times 10^{-9}$ bar $\times$ 0.1 mol/kg/bar = $0.05 \times 10^{-9}$ mol/kg $\approx$ 0.05 nM.

The calculation shows the detection limit for H2S by smell is on the order of sub-nanomolar range. Under such a low concentration, the statement "sulphide compounds were most likely not present" is still robust.

[1] Iowa State University Extension (May 2004). The Science of Smell Part 1: Odor perception and physiological response. PM 1963a. [2] NIST Chemistry WebBook,

SRD 69. https://webbook.nist.gov/cgi/cbook.cgi?ID=C7783064&Mask=10

[Referee] 9, 5: I would leave out this back-of-the-envelope estimate of denitrification. It does not add new, robust insight to N loss in C. B.

[Response] Indeed. We will remove this paragraph in the next version.

[Referee] 9, 23 and onwards: This is an important finding, which requires elaboration. I suggest calculating the direct contribution from nitrate to N2O for all the different combinations of nitrate and nitrite concentrations instead of just one example. If rates are assumed to be constant during the incubation, a simple model can describe the concomitant production and consumption of nitrite and hence how N-15 should accumulate in the extracellular nitrite pool if the intermediate nitrite were exchanging freely.

[Response] The reviewer points out the need to quantify intracellular nitrite exchange during nitrate reduction. We think it is impossible, and beyond the scope of this paper, to quantify the actual percentage of nitrite using the data presented here. We attempted to examine one hypothesis: nitrite is fully (100%) exchanged during nitrate reduction to N2O. And the calculation result shows that 15N-fraction labeled of N2O from the calculation does not match our measurements. Therefore, we reject the hypothesis. More elaborate experiments can be conducted in the future to tackle this question.

[Referee] 10, 9: I don't understand this formula. How can the amount of 15N-nitrite produced depend on either the total nitrate and the total nitrite concentration? Shouldn't it simply be: Rate of NO2- production from NO3- ×incubation time × initial fraction labelled of NO3-?

[Response] The formula was incorrect. Now the formula has changed to "Rate of NO2- production from NO3- × incubation time × initial fraction labelled of NO3- ". The result is 0.2 $\mu$mol-N L-1 hr-1 × 2 hr × 0.16 = 0.064 $\mu$mol-N L-1. And the revised value will be inserted to the subsequent calculations. The resulting 15N fraction of N2O will be 0.0087. This value is still much lower than measured value (>0.02) and our conclusion

is still robust.

[Referee] 12, 1: don't understand this. Oxygenation will just shift the zone of denitrification and N2O consumption to greater depth (until it reaches the sediment). It won't necessarily inhibit it.

[Response] Good point. The N2O concentration profile in July indicates that N2O consumption is occurring. Our incubation experiment showed N2O production is occurring at the oxic-anoxic interface. These results demonstrate denitrification is responsible for N2O production and consumption in different layers of water column. Whether the Chesapeake Bay is net N2O source or sink requires further evaluation, and will be one of the future research directions. We'll add these in the next version of the manuscript.
* * *

---

## Author Response (AR1)

**Response to reviewers**

**Anonymous Referee #1**

The manuscript by Qixing Ji and colleagues investigates the controls on nitrous oxide concentrations in Chesapeake Bay. It is a topic that the authors are very familiar with and this expertise is reflected in the experimental design investigating the effect of nutrients and oxygen on nitrous oxide. The datasets are useful and should be published, but I recommend restructuring the manuscript to focus on the strengths of the work and dealing with the issues raised.

Major comments

[Referee] As can happen with studies which conduct repeated experiments at different time intervals with varying measurements, it is difficult at times to track all the activity. I believe there is a discrepancy between the Methods and Results, as Figure 2 shows $N_2O$ production rates for July, November, and May, yet in the Methods you state that the experiments were only conducted in July. Instances like this make it very difficult for the reader to follow.

[Response] The $N_2O$ production rates shown in figure 2 are measured in control incubations, which were performed on all three dates. In control incubations, samples received 5 µmol $L^{-1}$ $^{15}N$-nitrate or $^{15}N$-nitrite, respectively; and oxygen was removed from samples by helium flushing. The DIN manipulation experiments were only conducted in July. We explained this in page 4 line 99 and revised table 1 to minimize confusion.

[Referee] I had a hard time relating the experiments with the estuarine environment. If you want to measure nitrous oxide emissions from Chesapeake Bay, the work conducted in this study is not what needs to be conducted. You would need high resolution surveys of key hydrographic parameters coupled with nitrous oxide measurements, either continuously or at targeted times. I recommend the authors focus more on the experiments as a means to better understand the controls on nitrous oxide production and use Chesapeake Bay as the contextual background, rather than attempting to explain nitrous oxide dynamics in this estuary.

[Response] We agree with the reviewer's suggestion, and this focus on control is exactly what we intended, with the title stating explicitly that this manuscript is about examination of the control of $N_2O$ production, rather than emissions, from the Chesapeake Bay. Previous study suggested the Chesapeake Bay is a $N_2O$ source, which motives our study (line $47 - 56$). We mention briefly that water column $N_2O$ the may be emitted due to disruption of water column stratification (page 12, line $291 - 294$), which points to further research directions. We'll revise accordingly in the next version.

[Referee] Two examples of the mis-match between the datasets reported and the contextual description provided in the Introduction are (1) The abstract talks about intense $N_2O$ efflux from estuaries, but the results show nitrous oxide concentrations close to air-saturation values. (2) The title mentions seasonal anoxia, but this is not shown in the results.

[Response] For (1), this is a more of a "general" statement that estuaries can be sites of intense $N_2O$ efflux, and previous studies have shown that the Chesapeake Bay is a $N_2O$ source. This context motivated our research. The measurements and experiments showed that despite the fact that the water column in July was close to $N_2O$ saturation level (line $171 - 175$), $N_2O$ production can occur (line $196 - 200$).

For (2) The oxygen profiles obtained from this study (and many others) show that water column anoxia occurs in summer. In spring and autumn, the water column is oxygenated. These results demonstrate the water column anoxia is a seasonal event, as clearly shown in the two papers by Lee et al., which are cited in the manuscript.

A revised abstract will be presented in the next version to minimize confusion to the readers.

[Referee] I am not sure if the May 2017 dataset is necessary in Figure 1. It is helpful in Figure 2 only.

[Response] The May dataset illustrates seasonal variation in the usual parameters, and thus supports the seasonal nature of $N_2O$ production and consumption. In May 2017, low oxygen condition was forming, along with higher availability of nitrate. These conditions are becoming favorable for denitrification in the water column.

[Referee] You need to include a description of how you calculate N2O production in the Methods section.

[Response] The rate calculation is presented in section 2.4, equation 1

[Referee] The $N_2O$ profiles puzzle me in the context of the other results. It looks like July 2016 is the only profile which has significant differences with depth, decreasing in concentration between 3 and 13 m. However, this time point is associated with the highest rates of $N_2O$ production (Figure 2). Therefore, $N_2O$ consumption is very important yet hardly mentioned in the manuscript. In context of your comment that estuaries are emitting large quantities of $N_2O$, the consumption processes deserve more attention.

[Response] $N_2O$ consumption is indeed important; however it was not quantified nor addressed directly in the current manuscript. The work here focuses on the controls of $N_2O$ production via denitrification. One of the future research direction is to examine the $N_2O$ reduction during anoxic events. The fact that significant $N_2O$ production was detected in the month when $N_2O$ concentration was lowest implies that $N_2O$ consumption was also occurring and probably minimizing efflux to the atmosphere (line 294 – 297).

[Referee] You should explain to the reader why you focused on the nirS gene and not other relevant genes

[Response] Good point. The nirS gene encodes the genetic material for nitrite reductase, the enzyme responsible for nitrite reduction to nitric oxide. NirS is often used as a proxy for the abundance and diversity of denitrifying bacteria (which was our application here) and is the gene in the denitrification sequence that is most reliably associated with a complete denitrification pathway (Graf et al. 2014). We explain in line 37 – 39.

Minor comments
[Referee] Page 1 Line 16 Change reducing to decreasing
[Response] Corrected.

[Referee] Page 2 Line 11 Agriculture such as paddy fields?
[Response] Is the reviewer referring to "agricultural land" in line 13? If so, paddy fields are not common in the Chesapeake Bay watershed. Agriculture is the main land use in this region and major agricultural activities in the area include livestock farming, greenhouse and nursery products (flowers, ornamental shrubs, and young fruit trees), corn and soybeans, all of which depend heavily on industrial fertilizers.

[Referee] Page 4 Oxidized nitrogen. I think this refers to NO3 and NO2. I recommend you write NO3 and NO2 if this is correct, as it avoids NO or NOx. Also on Page 7, Line 18
[Response] Corrected.

[Referee] Page 3 Line 21 It would help orientate the reader if you provide a short explanation for why you chose these 3 depths. For example, why did you only sample below the oxycline in anoxic waters and out of curiosity, why did you not compare anoxic and oxic?

[Response] As the stated in the introduction, the anoxic events in the Chesapeake Bay is of great environmental and economic concerns. One of the motivations of this work is to examine the control of $N_2O$ production under anoxic condition. We sampled oxygenated condition in May and November because anoxic conditions were not detected at the depths where they would have been expected during the seasonal anoxia, thus the oxic conditions were sampled for comparison and, not surprisingly, had lower rates of denitrification. It is likely that nitrification is important during oxygenated conditions. Future work more focus more on the oxic waters.

[Referee] Page 4 Line 14 Why do you inject N2O to detect N2O production? Is it an issue of detection limit? Or were you looking for N2O consumption?
[Response] It is an issue of detection limit of mass spectrometer, which requires > 2 nmol of nitrogen. See line 93 – 94.

[Referee] Page 4 Line 14 If you inject 1.2 nmol, how do you get 20 nmol $L^{-1}$? Presumably you verified these target concentrations on a few bottles and you should state the final concentrations achieved.
[Response] $N_2O$ concentration in the incubation bottles was estimated as follows: The incubation bottle is 60 ml in volume. Considering only 3 ml of headspace in the bottle, 90 – 95 % will dissolve in water phase (57 ml, 0.057 L) under experimentally relevant conditions. Therefore, 1.2 nmol $\times$ 0.9 $\div$ 0.057 L $\approx$ 20.0 nmol $L^{-1}$. $N_2O$ concentration was directly measured in the time course samples on the mass spec, so no assumptions were necessary in the actual rate calculations.

[Referee] Page 5 Line 1. Again, presumably you checked the final concentrations of oxygen against your target concentrations. I suspect you also did the air-equilibration at a single temperature which you should state.
[Response] Oxygen saturated site water was made by vigorously shaking an open-capped bottle with water collected at depth. Concentration was calculated using formula from Gordon and Garcia, 1992, at measured temperature (< 0.5 degC difference of in situ) and salinity. Oxygen concentrations were not measured directly in these experiments.  We have, however, used an optical sensor to measure concentrations directly in the same kinds of bottles in similar experiments and the agreement between estimated target concentration and measured is excellent. Thus it is not necessary to measure every bottle every time. See line 113 – 114.

[Referee] Page 5, Line 19 The alternative to injecting N2O standards into crimp-sealed vials is to air-equilibrate at controlled temperatures. This might be easier?
[Response] The measurement of $N_2O$ is using purge-and-trap technique, which analyze all of $N_2O$ contained in the vials by a mass spectrometer. Thus, the total amount of $N_2O$ is important, even the $N_2O$ standards may not be in equilibrium between the water and headspace.

[Referee] Page 7, Line 13 Where is the intense efflux that you mentioned in the abstract?
[Response] As explained earlier, this refers to a previous study documenting the Chesapeake Bay as a $N_2O$ source. Conditions in estuaries are highly variable, in both time and space, which is one of the motivations for investigating the control mechanisms on $N_2O$ production.  Failure to detect efflux at this time does not mean that intense efflux does not occur at this site at other times, or in other parts of the Bay.

[Referee] Page 7, Line 13 If you are going to talk about saturation, you have to provide the saturation value for each of the three sampling occasions.
[Response] The surface $N_2O$ saturation values in July, November and May were: 6.6, 10.4 and 12.0 nmol $L^{-1}$, respectively. These values have been included in page 7, line 171.

**Anonymous Referee #2**

General Comments:

[Referee] The authors have presented a well-designed and well-executed experiment on N2O production in Chesapeake Bay. While the experiments and data are worthy of publication, the manuscript itself needs major revisions before it is accepted for publication. In reading the manuscript the introduction and methods were written clearly and concisely, but the results and discussion needs a substantial reworking. The paragraphs jumped from one topic to the next and often I found the subtitled headings were not appropriate for the range of topics covered in the text below. In the abstract the authors suggest two potential impacts of what reducing N to the bay will have on N2O production. I think the paper is missing a more detailed explanation of how the experiments done lead to the conclusions they make, and they should expand on the specific details of these impacts that they suggest. I also think the authors could elaborate on the limitations of their study and what could be done in the future.

[Response] The reviewer's major criticism concerns the structure of results and discussion, which we will explain later. The main objective of this work is to examine the control of water column $N_2O$ production via denitrification in the Bay. The experiments were designed to suit our objective, by increasing the availability of oxygen and dissolved inorganic nitrogen through manipulation, as well as measuring $N_2O$ production in different seasons. In doing so we concluded that increasing DIN availability and decreasing oxygen availability could stimulate denitrification during summertime anoxia. In fall and spring the N2O production rates via denitrification were low, probably due to lack of anoxic conditions and concomitant low abundance of denitrifiers.

The major limitation of our study, like other tracer incubation studies in aquatic environments, is that the $N_2O$ production rates measured here should be treated as potential rates because the experimental conditions were not 100% identical to in situ environments (see line 204 – 206). Despite these artifacts, the conclusions that DIN and oxygen availabilities control $N_2O$ production during anoxic events are robust. The current work could be complemented by these future research directions, (1) measurement of nitrification rates in oxic waters and associated $N_2O$ production and nitrification genes; (2) measurement of N2O reduction rates within anoxic layer and characterization of $N_2O$ dynamics in the Chesapeake Bay; (3) Elucidation of intracellular nitrite exchange during nitrate reduction in natural waters; (4) Investigation of the physical, chemical and biological controls of N2O emission in the Chesapeake Bay.

See the revised "Conclusion and outlook" section that explained the above.

Minor Comments:

Abstract:

[Referee] Pg1 line15: "N2O production was positively correlated with the ratio of nitrate to nitrite concentrations." Please clarify if this was in situ concentrations or the nitrate and nitrite added to the incubation.

[Response] This statement describes $N_2O$ production rates under manipulated nitrate and nitrite concentrations. All of rates reported in the manuscript were measured using [15]N tracers, meaning the substrate concentrations (nitrate and nitrite) were higher than in situ levels. We revised the sentence in line 204 – 206.

[Referee] Pg1 line18-19: What do you mean by nitrogen deficiency? What lengths of time are your referring to when you write "short term" and "in the long-run", can you approximate the time scale?

[Response] The term "nitrogen deficiency" refers to nitrate and nitrite concentrations below detection. This occurs during 4 months of summer from June to September. We use "short-term" to describe a time scale of months, and "long-run" of years. See line 19.

Methods:
[Referee] Pg4 line15: Why do you add 20 nM N2O before the incubation? If it is so you can have enough N2O to measure the N2O production, could you add it after the incubation is killed? Can you clarify here in the text? Do you think adding it before, could affect the production of N2O?
[Response] It is an issue of detection limit of mass spectrometer, which requires > 2 nmol of N (see line 93 – 94). We agree that the added tracer concentration is slightly higher than in situ (6 – 12 nM). It is our intention to add the N$_2$O before the samples were preserved so as to have a more similar condition to in situ. The effect of N2O concentration on the rate of N2O production should be investigated.

[Referee] Pg4 line17: What was the total concentration of nitrite+nitrate of tracer added?
[Response] Here is the description of DIN manipulation experiment, which was conducted in July 2016 only (see line 99 – 100). The total concentrations of nitrate and nitrite were 5 μM each, and all the substrate concentrations of every experiments conducted are listed in table 1.

[Referee] Pg5 line8: How does 5uM of tracer compare to the in situ amounts of nitrite and nitrate.
[Response] This was stated in page 7. In July 2016, in situ nitrate and nitrite concentrations were below detection (0.02 μM), and the experimental conditions are in no way representing the in situ. Hence we treated the rates as potential rates. In November 2016, in situ nitrate and nitrite concentrations at the depth of experiment (17 m) were 5.0 and 0.4 μM, respectively.

Section 3.2 Active N2O production by denitrification
[Referee] Pg8 line3: In the title you suggest all N2O is from denitrification, it would be good discuss why it is not from nitrification, if you are using that as the section title.
[Response] Indeed. We only measured N$_2$O production via denitrification. Unfortunately we did not measure N$_2$O production via nitrification. We changed the title to "Active N$_2$O production in the water column" to minimize confusion.

[Referee] Pg8 line16: I think it's good you say here they are potential rates of N2O production, because you are removing all the oxygen, even if that's not the in situ value. I am curious why there are no potential rates in May. Can you add some discussion on why removing the oxygen does not stimulate N2O production in May versus November?
[Response] We explained in the following paragraph (see line 209 – 217). This is likely due to low denitrifier abundance (indicated by nirS gene abundance) in May, during which the anoxic condition had not yet developed. In addition, our anoxic incubation lasted around 2 hours; it is unlikely for denitrifiers to reactivate within such a short time scale.

[Referee] Pg9 lines1-13: This section seems disjointed from the paragraph above and the title of the section. This paragraph refers more to removal of fixed nitrogen from the bay rather than N2O production.
[Response] Indeed. We will remove this paragraph in the next version.

Section 3.3 N2O production pathways regulated by availability of nitrogen substrate

[Referee] Pg9 line16: I was confused by this term "NO2- (NO3-), is this synonymous with "NO2- or NO3- "? If so, I would suggest changing it to the latter.

[Response] This is simply a combination of two sentences in one: "… increasing nitrite availability favors $N_2O$ production from nitrite reduction" and "… increasing nitrate availability favors $N_2O$ production from nitrate reduction." We change the sentence to "This suggests increasing $NO_2^-$ or $NO_3^-$ availability favors $N_2O$ production from the reduction of respective substrate." in line 220 – 221.

[Referee] Pg10 lines1-21, pg 11 lines1-8: This could use a different subheading? Could you reformat the equations to be easier to read? Also in general this section could use a little more description, as is, it is a little hard to follow. I would start off the section with the punch line (on pg 11) and then describe why the calculations back up that statement.
[Response] We think another sub-heading is probably not necessary but we will reorganize the section in the next version. We'll add the conclusion "the exchange between intracellular and ambient nitrite during nitrate reduction to $N_2O$ is limited" in the beginning of the section. The calculation is to find the $^{15}N$ fraction label of $N_2O$ when nitrite is fully exchanged, and we think it is rather straightforward. We used a different font and size to better illustrate the calculation in line 232 – 248.

Section 3.4 Oxygen inhibits N2O production by denitrification

[Referee] Title: I suggest changing the title to something more detailed.
[Response] The title states the major conclusion of this section and seems pretty specific.

[Referee] Pg11 lines11-19: These sentences seem to belong to the other section about N influx to the bay. They could be moved or deleted during restructuring of discussion sections because it is somewhat repetitive from before.
[Response] Indeed. This paragraph will be organized in the "conclusion and outlook" section.

[Referee] Pg12 lines9-15: The result that there is a different oxygen tolerance for nitrite vs. nitrate reduction is really interesting. I think here is where you should expand on more why you get this result. I know you touch on it more in a later section, but it might be best to put it all together (maybe in it's own section). Could it be from nitrifier denitrification? How would nitrite oxidation affect the production rate? At low levels of oxygen I would suspect there would be some nitrite oxidation to nitrate.
[Response] It is possible that nitrifier denitrification is responsible for part of N2O production via nitrite reduction. This is because increasing oxygen availability will inhibit the activity of denitrifiers, but not nitrifiers (see line 277 – 285). It is not possible to have a complete explanation with the data presented here (see line 150 – 152).
We did not measure nitrite oxidation rates in the samples but agree that it is likely nitrite oxidation was also occurring. We expect nitrite oxidation to have a negligible effect on N2O production rates, however, since N2O is not a product or intermediate in that reaction and expected rates of nitrite oxidation would not significanty affect the concentration of the substrates directly involved in N2O production. (Nitrite oxidations rates in nM/d would not affect the substrate concentrations, which were in the micromolar range.)

[Referee] Pg12 lines16-19: This paragraph seems out of place? Again, would you suspect some nitrite oxidation in these nitrate reduction to nitrite measurements? How would that affect results?
[Response] See above.

[Referee] Pg13 lines3-4: This line should go with the section on differences of nitrite-stimulated production vs. nitrate-stimulated N2O production.

Pg13 lines10-18: This paragraph is out of place.
Pg13 lines19-22: Should this be in the conclusion and outlook?
[Response] The section 3.4 was re-organized to improve clarity and make tighter connections between the data and conclusions.

Section 4 Conclusion and outlook
Pg14 lines14-19: I would reorder this section and put these lines first.
Conclusion and outlook section as a whole: Could you add something connecting the two pathways of management (nitrogen influx vs. oxygenation)? Also, could you quantify how each change inhibits N2O production?

[Response] Indeed. The current work will be complemented by these future research directions, (1) measurement of nitrification rates in oxic waters and associated $N_2O$ production and nitrification genes; (2) measurement of N2O reduction rates within anoxic layer and characterization of $N_2O$ dynamics in the Chesapeake Bay; (3) Elucidation of intracellular nitrite exchange during nitrate reduction in natural waters; (4) Investigation of the physical, chemical and biological controls of N2O emission in the Chesapeake Bay.

See "Conclusion and outlook" section that explained the above.

**Anonymous Referee #3**

Summary
The authors present an examination of N2O dynamics studied in the Chesapeake Bay during three samplings. Since the Chesapeake Bay exhibits a strong seasonal shift in water column redox state, the study focused on trying to link these shifts with N2O production mechanisms at and below the oxic/anoxic interface. The authors bring a range of chemical, molecular and isotopic tools to bear on these dynamics, with an emphasis on elucidating N2O producing processes occurring in the bottom waters and the primary controls on them. This contribution is timely – as coastal and estuarine systems are dynamic and generally understudied with respect to their place in the global N2O budget. Overall the data appear to be of high quality. The manuscript is generally well written, though parts could benefit from some reorganization. I have some questions about the data interpretation as outlined below. Overall I think this work is worthy of publication, but that the manuscript could be improved through some more careful consideration of clarifying some sections.

Major Comments
[Referee] Pg 9 Ln 15: I appreciate the use of targeted assays for $NO_2^-$ and $NO_3^-$ reduction, though it is unclear whether this was designed to constrain/target nitrifier denitrification specifically or explicit nitrite reducing denitrifiers (???). Clearly denitrifying organisms also use NO2- in their electron transport chain. In Section 3.3 the authors attempt to tackle this – and I appreciate the argument that they are making about NO2- transport across the membrane – but I feel that this section is confusing as written. Using calculations laid out here, and making a few key assumptions, the authors conclude that since the level of $^{15}N$ label in the N2O pool is much higher than if there had been full exchange, then exchange between the cellular and ambient NO2- is minimal. I acknowledge that this is a difficult aspect of N cycling to track, but I am not overly convinced that they have proven that this type of exchange is 'minimal.' Their calculation demonstrates that high levels of exchange are not occurring, but whether modest levels might be influencing the results is unclear. Perhaps this argument could be streamlined and clarified.
[Response] The reviewer acknowledges the difficulty of our attempt to estimate intracellular nitrite exchange; we appreciate it. The hypothesis is: nitrite is fully (100%) exchanged during nitrate reduction to $N_2O$. And the calculation result shows that 15N-fraction labeled of N2O from the calculation does not match our measurements (see line 249). Therefore, we reject the hypothesis. Yes, it is possible that some level of exchange might occur, but they would be undetectable by this argument. We think it is impossible, and beyond the scope of this paper, to quantify the actual percentage of nitrite using the data presented here. More elaborate experiments can be conducted in the future to tackle this question.

[Referee] Additionally, perhaps the introduction needs some sort of clearer description of the types of metabolisms being targeted by the study (complete denitrifiers, nitrite denitrifiers, nitrifier denitrification). These classifications of microbes and processes are confusing even to those who regularly study N cycling.
[Response] We focus only the reductive pathways of $N_2O$ production under anoxic conditions. Given the limitation of $^{15}N$ tracer study, only the denitrification pathways "$NO_2^- \rightarrow N_2O$" and "$NO_3^- \rightarrow N_2O$" can be quantified (line 59 – 60). It is therefore very difficult to attribute the pathway to certain groups of microbes. To minimize confusion, we generalize the term "$N_2O$ production during denitrification" throughout the text. The current data set does not support in-depth discussion of functional microbial groups responsible for $N_2O$ production because we cannot differentiate among the different kinds of microbes that can perform nitrite reduction to nitrous oxide using tracer experiments.

Minor Comments

[Referee] Pg 1 Ln15: I believe nitrate and nitrite are reversed here (and many other times throughout – leading to some frustration/confusion).

[Response] We did not follow the reviewer's suggestion for this sentence. The experimental data shows that, higher nitrate or nitrite availability positively correlates with $N_2O$ production rates from respective substrates. The sentence itself is correct.

[Referee] Pg 1 Ln17: Since the field data demonstrate that there is no net flux to the atmosphere– it seems odd to emphasize N2O efflux here.

[Response] To minimize confusion, we changed the word "efflux" to "production"

[Referee] Pg 3 Ln 19: Please clarify whether a headspace was left in the incubation bottle or not.

[Response] The headspace (3 mL) was left throughout the incubation.

[Referee] Pg 4 Ln 1: course not courses

[Response] Thanks for catching that error – it was in line 21.

[Referee] Pg 5 Ln 3: Please indicate whether oxygen concentrations were measured or calculated?

[Response] The concentrations were calculated, see line 110 – 113.

[Referee] Pg 6 Ln 1: Why were no other functional gene assays performed? I think this is justified later, but given that nirS only reflects denitrification – its linkage with N2O from this pathway is clear, yet reveals little about the dynamics of the other pathways investigated as I understand. Why not include nirK? Or norB?

[Response] Good point. The nirS gene encodes the genetic material for nitrite reductase, the enzyme responsible for nitrite reduction to nitric oxide. NirS is often used as a proxy for the abundance and diversity of denitrifying bacteria (which was our application here) and is the gene in the denitrification sequence that is most reliably associated with a complete denitrification pathway (Graf et al. 2014). See line 37 – 39.

[Referee] Pg 7 Ln 19: I believe nitrate and nitrite are reversed here again.

[Response] Corrected.

[Referee] Pg 7 Ln 25: "positively correlates" – yes, but this is difficult to defend statistically with n=3.

[Response] The sentence is changed to "…nirS abundance increases with increasing measured rates of $N_2O$ production." See line 192.

[Referee] Pg 13 Ln 2: I would suggest "microbial groups" instead of microbial communities (which may imply the 'greater community' – not just N cycling organisms ?).

[Response] The word "groups" has replaced "commnuities".

[Referee] Pg 14 Ln 5: It seems that if nitrifier denitrification and ammonia oxidation are implicated in N2O production as discussed – then the nitrifier community dynamics would also play an important role and should be acknowledged?

[Response] Indeed. The reviewer points out one of the future research direction, that is to examine the link between nitrifying community and $N_2O$ production via nitrification. See line 310 – 317.

**Anonymous Referee #4**

[Referee] The paper reports an experimental study of rates and pathways of nitrous oxide production in Chesapeake Bay waters. Water was sampled on three occasions (spring, summer, autumn) and incubated with N-15 labelled nitrate or nitrite under anoxic conditions. Additional incubations were made with oxygen added back to investigate the oxygen sensitivity of the processes. Based on the results, the authors draw conclusions about the controls of N2O emissions from the Bay. The paper addresses an interesting subject and the experimental work is of good quality. However, the results do not provide strong support for the conclusions because the experimental conditions do not sufficiently reflect the environmental conditions in the Bay.

Also, although there are few previous experimental studies from comparable environments, the paper largely neglects the large number of previous studies on N2O dynamics in estuaries, although these do provide some insight to the controls of N2O emission.

Importantly, the literature points to nitrification (ammonium oxidation) as a major $N_2O$ source in estuaries whereas the present study only investigates N2O production through denitrification. Without data on the rates and controls of N2O production by ammonia oxidation (i.e. experiments with N-15 labelled ammonia at different oxygen concentrations), no conclusions can be drawn about the controls of N2O emissions from Chesapeake Bay.

Based on the mismatch between the experiments and the conclusions, I recommend that the paper be rewritten to focus on what the experiments can actually tell us, i.e. how N2O production during denitrification is affected by oxygen, and how production from nitrate and nitrite seem to function independently, which is novel and interesting.

I also warn against trying to translate the results into understanding denitrification as a N2O source in the Bay as a whole, and the role of anoxia in this, because denitrification is an interface process there, and the anoxic water body might serve as net sink for N2O, drawing it down from the overlying oxycline. Here, fine scale profiling of N2O across the interface might be more informative than the experimental approach.

[Response] The reviewer's major criticism is that the manipulated experimental conditions led to the conclusion of the Chesapeake Bay being a $N_2O$ source, whereas the static concentration profile suggested the Bay as a $N_2O$ sink. We shall explain as follows:

(1) Our work of environmental control of $N_2O$ production is motivated by previous studies that identified the Chesapeake Bay as a $N_2O$ source (Elkins et al., 1978; McElroy et al., 1978).

(2) Through incubation experiments, it is straightforward to draw the conclusion from the results: Adding nitrogen substrates and removing oxygen stimulate $N_2O$ production in summer and autumn. Thus the Bay is potentially a $N_2O$ source when pulses of nitrogen enters the water body that is experiencing summertime anoxia (see line 204 – 208). Conditions in estuaries are highly variable, in both time and space, which is one of the motivations for investigating the control mechanisms on $N_2O$ production. Failure to detect efflux at this time does not mean that intense efflux does not occur at this site at other times, or in other parts of the Bay.

(3) Indeed there's a large body of literature reporting the variation of $N_2O$ fluxes in estuaries around the world. Many of them appleid the traditional methodology: By quoting surface $N_2O$ supersaturation, wind speed, temperature, water turbulence, etc. and measure $N_2O$ fluxes, which is then related to water column oxygen and nitrogen availability. A small number of work applied nitrogen isotopic approach to study the mechanism of $N_2O$ production. The isotopic approach reveals the dynamic, potential $N_2O$ production that is masked by the static

concentration profile. In addition, the isotopic approach allows to quantify the environmental controls of $N_2O$ production in the water column.

We disagree with the reviewer about the lack of similarity between experimental and actual conditions. Our incubation experiments were designed to study the effects of oxygen and nitrogen availability on $N_2O$ production, by changing experimental nitrogen $(1 - 10 \, \mu M)$ and oxygen concentrations $(0 - 10 \, \mu M)$ that often occur in the Chesapeake Bay (Lee et al., 2015).

As we have stated in the Introduction, nitrification is another important pathway for $N_2O$ production. The focus of the manuscript is $N_2O$ production under naturally occurring and laboratory anoxic condition. It is unlikely that nitrification could occur, and thus nitrification is beyond the scope of this manuscript. We thank the reviewer for pointing out one possible future research direction: study the $N_2O$ production via nitrification and denitrification and their environmental controls with extended spatial and temporal coverage across the Chesapeake Bay. See line $310 - 317$.

The scope of this manuscript is to examine the control of $N_2O$ production by nitrogen and oxygen availability during denitrification under naturally occurring and laboratory anoxic condition. It is another future research direction that study the $N_2O$ dynamics by fine scale profiling of $N_2O$ across the interface so as to demonstrate whether the anoxic Chesapeake Bay serves as a net sink or source for $N_2O$.

Specific comments
[Referee] 3, 3: Why pilot? – to me this indicates preliminary results
[Response] We chose "pilot" because we are the first group to study $N_2O$ production using 15N tracer at a single station in the Chesapeake Bay, and we examined the environmental control of denitrification pathway for $N_2O$ production. This is a small scale study and the results are important for a larger scale, more comprehensive study in the future.

[Referee] 7, 10: The detection limit for H2S by smell is~ $10 \mu M$. It seems strange if no sulphide was present at all, if all more favourable oxidants were depleted.
[Response] It would be more helpful if the reviewer list the reference.
First, the odor threshold of $H_2S$ is $\sim 0.5$ ppb [1], and Henry's Law constant of $H_2S$ is $\sim 0.1$ mol/kg/bar [2]. Under atmospheric pressure with 0.5 ppb $H_2S$, the equilibrium concentration of a $H_2S$ solution is
$0.5 \times 10^{-9}$ bar $\times 0.1$ mol/kg/bar $= 0.05 \times 10^{-9}$ mol/kg $\approx 0.05$ nM.

The calculation shows the detection limit for $H_2S$ by smell is on the order of sub-nM range. Under such a low concentration, the statement "sulphide compounds were most likely not present" still holds.

[1] Iowa State University Extension (May 2004). The Science of Smell Part 1: Odor perception and physiological response. PM 1963a.
[2] NIST Chemistry WebBook, SRD 69.
https://webbook.nist.gov/cgi/cbook.cgi?ID=C7783064&Mask=10

[Referee] 9, 5: I would leave out this back-of-the-envelope estimate of denitrification. It does not add new, robust insight to N loss in C. B.
[Response] Indeed. We will remove this paragraph in the next version.

[Referee] 9, 23 and onwards: This is an important finding, which requires elaboration. I suggest calculating the direct contribution from nitrate to N2O for all the different combinations of nitrate and nitrite concentrations instead of just one example. If rates are assumed to be constant

during the incubation, a simple model can describe the concomitant production and consumption of nitrite and hence how N-15 should accumulate in the extracellular nitrite pool if the intermediate nitrite were exchanging freely.

[Response] The reviewer points out the need to quantify intracellular nitrite exchange during nitrate reduction. We think it is impossible, and beyond the scope of this paper, to quantify the actual percentage of nitrite using the data presented here. We attempted to examine one hypothesis: nitrite is fully (100%) exchanged during nitrate reduction to $N_2O$. And the calculation result shows that 15N-fraction labeled of $N_2O$ from the calculation does not match our measurements (see line 249). Therefore, we reject the hypothesis. More elaborate experiments can be conducted in the future to tackle this question.

[Referee] 10, 9: I don't understand this formula. How can the amount of 15N-nitrite produced depend on either the total nitrate and the total nitrite concentration? Shouldn't it simply be: Rate of $NO_2^-$ production from $NO_3^-$ $\times$ incubation time $\times$ initial fraction labelled of $NO_3^-$?

[Response] The formula was incorrect. Now the formula has changed to "Rate of $NO_2^-$ production from $NO_3$- $\times$ incubation time $\times$ initial fraction labelled of $NO_3^-$". The result is 0.2 µmol-N $L^{-1}$ $hr^{-1}$ $\times$ 2 hr $\times$ 0.16 = 0.064 µmol-N $L^{-1}$. And the revised value will be inserted to the subsequent calculations. The resulting $^{15}N$ fraction of $N_2O$ will be 0.0052 (see line 248).

[Referee] 12, 1: don't understand this. Oxygenation will just shift the zone of denitrification and N2O consumption to greater depth (until it reaches the sediment). It won't necessarily inhibit it.

[Response] Good point. The $N_2O$ concentration profile in July indicates that $N_2O$ consumption is occurring. Our incubation experiment showed $N_2O$ production is occurring at the oxic-anoxic interface. These results demonstrate denitrification is responsible for $N_2O$ production and consumption in different layers of water column. Whether the Chesapeake Bay is net $N_2O$ source or sink requires further evaluation, and will be one of the future research directions. See line 287 – 299.

---

## Author Response (AR2)

Response to Reviewer

*Italic: reviewer's comments*
Underline: Authors' response

*The authors refute my major concerns about the paper. They have, however, made changes in response to the other reviewers' comments that partly resolve these issues.*

*My major concern was that while the study only examines one pathway of N2O production, denitrification, and pathways associated to nitrification are ignored, the text in several places seems to deal N2O production in general, and not just one of the pathways (e.g., abstract l. 11, 12, 16, 18, 20; l. 273). The authors in their response argue that nitrification is not important during anoxia, which is obvious, but it is not obvious to the reader of the abstract that only (or mainly) anoxic experiments were conducted and that only denitrifying pathways were quantified. This must be made clear from the beginning, e.g., l. 11 should be changed to "…were used to investigate the geochemical factors controlling N2O production FROM DENITRIFICATION in the Chesapeake Bay", and so on for the other places mentioned above.*

We made revisions in line 8 – 9, 10 – 12 in the abstract, and line 274 – 275 in discussion section, plus the statement in line 59 – 61 to emphasize that the main focus of this pilot study is quantifying the $N_2O$ production rates from denitrification and associated geochemical factors. We acknowledged in line 313 – 320 that, nitrification is also another important production pathway that awaits further research effort.

The authors also refute my comment about the detection for H2S by smell being ~10 µM. They do so, however, by referring to a naïve calculation, which has little to do with the context in which (I guess) the olfactory assay was conducted (on deck, in turbulent air, placing a nose above a bottle or stream of water, and not by equilibrating a large volume of water and inhaling the equilibrated air purely – if I'm wrong this needs to be specified). Thresholds in the low µM range can be found as a rule of thumb in the scientific literature and on the www – I leave the search to the authors. The point here is, however, that the conclusion that H2S was absent is not justified.

We revised the statement in line 169 – 170 to adopt the reviewer's suggestion: "The water samples were free of any hydrogen sulphide odor, so we conclude that sulphide was either absent or was present at very low level (< 1 µmol L$^{-1}$)."

*The first version contained a very rough calculation of nitrogen removal by denitrification based on the N2O production rates. In their response, the authors say that it will be removed. Now I find it in the conclusions (l. 300-9). This is confusing.*

We consider this evaluation useful in that $N_2O$ production via denitrification could indicate the effectiveness of nitrogen removal by the sediment-water system. Thus nitrogen removal in estuaries comes with a small cost: emitting $N_2O$ as a by-product. This also points out an interesting research characterizing a potential negative feedback: "increase nitrogen loading → deoxygenation → nitrogen removal → decrease nitrogen loading". We revised the statement in line 310 – 312 to preserve our intent.